# Improved Imaging by Invex Regularizers with Global Optima Guarantees

**Samuel Pinilla[1,2], Tingting Mu[3], Neil Bourne[2], Jeyan Thiyagalingam[1]**

[1][*] Scientific Computing Department, Science and Technology Facilities Council, Harwell, UK
[2]University of Manchester at Harwell, UK
[3]Computer Science, University of Manchester, UK
{samuel.pinilla,t.jeyan}@stfc.ac.uk
{tingting.mu,neil.bourne}@manchester.ac.uk

## Abstract

Image reconstruction enhanced by regularizers, e.g., to enforce sparsity, low rank or smoothness priors on images, has many successful applications in vision tasks such as computer photography, biomedical and spectral imaging. It has been well accepted that non-convex regularizers normally perform better than convex ones in terms of the reconstruction quality. But their convergence analysis is only established to a critical point, rather than the global optima. To mitigate the loss of guarantees for global optima, we propose to apply the concept of *invexity* and provide the first list of proved invex regularizers for improving image reconstruction. Moreover, we establish convergence guarantees to global optima for various advanced image reconstruction techniques after being improved by such invex regularization. To the best of our knowledge, this is the first practical work applying invex regularization to improve imaging with global optima guarantees. To demonstrate the effectiveness of invex regularization, numerical experiments are conducted for various imaging tasks using benchmark datasets.

## 1 Introduction

Image reconstruction (restoration) enhanced by regularizers has a wide application in vision tasks such as computed tomography [1, 2], optical imaging [3, 4], magnetic resonance imaging [5, 6], computer photography [7, 8], biomedical and spectral imaging [9, 10]. In general, an image reconstruction task can be formulated as the solution of the following optimization problem:

$$\underset{\boldsymbol{x}\in\mathbb{R}^n}{\text{minimize}} \quad F(\boldsymbol{x}) = f(\boldsymbol{x}) + g(\boldsymbol{x}). \tag{1}$$

Here $f(\boldsymbol{x})$ models a data fidelity term, which usually corresponds to an error loss for image reconstruction, and is assumed to be differentiable. The other function $g(\boldsymbol{x})$ acts as a regularizer which can be non-smooth. It imposes image priors such as sparsity, low rank or smoothness [11]. The use of an appropriate regularizer plays an important role in obtaining robust reconstruction results.

Convex regularization has been popular in the last decade [11, 12, 13, 14, 15], because it can result in guaranteed global optima. The most well-known examples include the $\ell_1$-norm and nuclear norm, which are the continuous and convex surrogates of the $\ell_0$-pseudo norm and rank, respectively [16]. Although convex regularizers have demonstrated their success in signal/image processing, biomedical informatics and computer vision applications [13, 17, 18, 19], they are suboptimal in many cases, as they promote sparsity and low rank only under very limited conditions (more measurements from the scene are needed [20, 21]). To address such limitations, non-convex regularizers have been proposed.

---

[*]Rutherford Appleton Laboratory.

36th Conference on Neural Information Processing Systems (NeurIPS 2022).

Table 1: Comparison between the assumptions made in this work for $f(\boldsymbol{x})$, and $g(\boldsymbol{x})$ to be optimized in Eq. (1) and the most common/successful assumptions in the state-of-the-art.

| Method name | Assumption | Global optimizer |
|---|---|---|
| IRLS [33, 34] | special $f$ and $g$ | No |
| General descent [35, 36] | Kurdyka-Łojasiewicz | No |
| GIST [37] | nonconvex $f$, $g = g_1 - g_2$, $g_1, g_2$ convex | No |
| iPiano [38] | nonconvex $f$, convex $g$ | No |
| **Proposed** | convex $f$, invex $g$ | **Yes** |

For instance, several interpolations between the $\ell_0$-pseudonorm and the $\ell_1$-norm have been explored including the $\ell_p$-quasinorms (where $0 < p < 1$) [22], Capped-$\ell_1$ penalty [23], Log-Sum Penalty [20], Minimax Concave Penalty [24], Geman Penalty [25]. However, these non-convex regularizers unfortunately come with the price of losing global optima guarantees.

Image reconstruction methods based on Eq. (1) include model-based approaches that directly solve Eq. (1) using well-established optimization techniques, e.g., proximal operators and gradient descent rules [26, 27, 28], learning-based approaches that train an inference neural network [29, 30], as well as hybrid approaches that draw links between iterative signal processing algorithms and the layer-wise neural network architectures [31, 32]. Many of these exploit non-convex assumptions over $f(\boldsymbol{x})$ and/or $g(\boldsymbol{x})$, for which we present a summary of some commonly used or successful ones in Table 1. The table includes algorithms like the iterative reweighted least squares (IRLS) [33, 34], where the regularizer is a composition between the one-dimensional $\ell_p$-quasinorm and the trace of a matrix. In [35, 36], the objective function $F(\boldsymbol{x})$ is assumed to form a semi-algebraic or tame optimization problem solved by gradient descent algorithms. In [37], the regularizer $g(\boldsymbol{x})$ is assumed to be the subtraction of two convex functions, and the general iterative shrinkage and thresholding (GIST) algorithm is proposed to optimize $F(\boldsymbol{x})$. Lastly, [38] assumes non-convex $f(\boldsymbol{x})$ but convex $g(\boldsymbol{x})$ and proposes the inertial proximal (iPiano) algorithm for optimization.

For algorithms with the convexity assumptions removed, e.g., those in Table 1, their convergence analysis unfortunately can only be established for a critical point. Ideally, we always prefer algorithms that can find the optimal solution for the target problem. One way to mitigate the loss of guarantees for global optima is by revisiting the concept of *invexity* which was first introduced by Hanson [39], Craven and Glover [40] in the 1980s. What makes this class of functions special is that, for any point where the derivative of a function vanishes (stationary point), it is a global minimizer of the function. Convexity is a special case of invexity. Since 1990s, a lot of mathematical implications for invex functions have been developed, but with the lack of practical applications [41]. Examples of the few successful works implementing the invexity theory include [42, 43, 44]. To the best of our knowledge, there is no existing work on the application of invex regularization for imaging.

In this paper, we focus on image reconstruction problems formulated in the form of Eq. (1), where the data fidelity term $f(\boldsymbol{x})$ is based on the $\ell_2$-norm and an invex regularizer $g(\boldsymbol{x})$ is used. Most invex theory research lacks clarity on how to benefit practical applications, and this does not encourage the practitioners to exploit the invex property [41]. We aim at filling this gap by providing for the first time concrete and useful invex optimization formulations for imaging applications.

Specifically, we make the following contribution:

- Provide the first list of regularizers with proved invexity that fits optimization problems for imaging applications.
- Establish convergence guarantees to global optima for three types of advanced image reconstruction techniques enhanced by invex reguarlizers.
- Empirically demonstrate the effectiveness of invex regularization for various imaging tasks.

## 2 Preliminaries

Throughout this paper, we use boldface lowercase and uppercase letters for vectors and matrices, respectively. The $i$-th entry of a vector $\boldsymbol{w}$, is $\boldsymbol{w}[i]$. For vectors, $\|\boldsymbol{w}\|_p$ is the $\ell_p$-norm. An open ball is defined as $B(\boldsymbol{x}; r) = \{\boldsymbol{y} \in \mathbb{R}^n : \|\boldsymbol{y} - \boldsymbol{x}\|_2 < r\}$. The operation $\mathrm{conv}(\mathcal{A})$ represents the convex hull of the set $\mathcal{A}$, and the operation $\mathrm{sign}(w)$ returns the sign of $w$. We use $\sigma_i(\boldsymbol{W})$ to denote the $i$-th singular value of $\boldsymbol{W}$ assumed in descending order.

We present several concepts needed for the development of this paper starting with the definition of a locally Lipschitz continuous function.

**Definition 1** (**Locally Lipschitz Continuity**). A function $f : \mathbb{R}^n \to \mathbb{R}$ is locally Lipschitz continuous at a point $\boldsymbol{x} \in \mathbb{R}^n$ if there exist scalars $K > 0$ and $\epsilon > 0$ such that

$$|f(\boldsymbol{y}) - f(\boldsymbol{z})| \leq K \|\boldsymbol{y} - \boldsymbol{z}\|_2, \tag{2}$$

for all $\boldsymbol{y}, \boldsymbol{z} \in B(\boldsymbol{x}, \epsilon)$.

Since the ordinary directional derivative being the most important tool in optimization does not necessarily exist for locally Lipschitz continuous functions, it is required to introduce the concept of subdifferential [45] which is calculated in practice as follows.

**Theorem 1** (**Subdifferential**). [45, Theorem 3.9] Let $f : \mathbb{R}^n \to \mathbb{R}$ be a locally Lipschitz continuous function at $\boldsymbol{x} \in \mathbb{R}^n$, and define $\Omega_f = \{\boldsymbol{x} \in \mathbb{R}^n | \ f$ is not differentiable at the point $\boldsymbol{x}\}$. Then the subdifferential of $f$ is given by

$$\partial f(\boldsymbol{x}) = \text{ conv } \left(\{\boldsymbol{\zeta} \in \mathbb{R}^n | \text{ exists } (\boldsymbol{x}_i) \in \mathbb{R}^n \setminus \Omega_f \text{ such that } \boldsymbol{x}_i \to \boldsymbol{x} \text{ and } \nabla f(\boldsymbol{x}_i) \to \boldsymbol{\zeta}\}\right). \tag{3}$$

The notion of subdifferential is given for locally Lipschitz continuous functions because it is always nonempty [45, Theorem 3.3]. Based on these, the concept of invex function is presented as follows.

**Definition 2** (**Invexity**). Let $f : \mathbb{R}^n \to \mathbb{R}$ be locally Lipschitz; then $f$ is invex if there exists a function $\eta : \mathbb{R}^n \times \mathbb{R}^n \to \mathbb{R}^n$ such that

$$f(\boldsymbol{x}) - f(\boldsymbol{y}) \geq \boldsymbol{\zeta}^T \eta(\boldsymbol{x}, \boldsymbol{y}), \tag{4}$$

$\forall \boldsymbol{x}, \boldsymbol{y} \in \mathbb{R}^n, \forall \boldsymbol{\zeta} \in \partial f(\boldsymbol{y})$.

It is well known that a convex function simply satisfies this definition for $\eta(\boldsymbol{x}, \boldsymbol{y}) = \boldsymbol{x} - \boldsymbol{y}$.

The following classical theorem [46, Theorem 4.33] makes connection between an invex function and its well-known optimum property that supports the motivation of designing invex regularizers.

**Theorem 2** (**Invex Optimality**). [46, Theorem 4.33]) Let $f : \mathbb{R}^n \to \mathbb{R}$ be locally Lipschitz. Then the following statements are equivalent.

1. $f$ is invex.

2. Every point $\boldsymbol{y} \in \mathbb{R}^n$ that satisfies $\boldsymbol{0} \in \partial f(\boldsymbol{y})$ is a global minimizer of $f$.

3. Definition 2 is satisfied for $\eta : \mathbb{R}^n \times \mathbb{R}^n \to \mathbb{R}^n$ given by

$$\eta(\boldsymbol{x}, \boldsymbol{y}) = \begin{cases} \boldsymbol{0} & f(\boldsymbol{x}) \geq f(\boldsymbol{y}), \\ \frac{f(\boldsymbol{x}) - f(\boldsymbol{y})}{\|\boldsymbol{\zeta}_{\boldsymbol{y}}^*\|_2^2} \boldsymbol{\zeta}_{\boldsymbol{y}}^* & \text{otherwise}, \end{cases} \tag{5}$$

where $\boldsymbol{\zeta}_{\boldsymbol{y}}^*$ is an element in $\partial f(\boldsymbol{y})$ of minimum norm.

## 3 Invex Functions

We start this section by firstly presenting five examples of invex functions that are useful for imaging applications. Four of these have been labelled as non-convex in existing works [47, 48]. This is the first time that they are formally proved to be invex functions. We prove their invexity by showing they satisfy Statement 2 of Theorem 2 (see proof in Appendix A of supplementary material).

**Lemma 1** (**Invex Functions**). All of the following functions are invex:

$$g(\boldsymbol{x}) = \sum_{i=1}^{n} \left(|\boldsymbol{x}[i]| + \epsilon\right)^p, \text{ for } p \in (0, 1) \text{ and } \epsilon \geq (p(1-p))^{\frac{1}{2-p}}, \tag{6}$$

$$g(\boldsymbol{x}) = \sum_{i=1}^{n} \log(1 + |\boldsymbol{x}[i]|), \tag{7}$$

$$g(\boldsymbol{x}) = \sum_{i=1}^{n} \frac{|\boldsymbol{x}[i]|}{2 + 2|\boldsymbol{x}[i]|}, \tag{8}$$

$$g(\boldsymbol{x}) = \sum_{i=1}^{n} \frac{\boldsymbol{x}^2[i]}{1 + \boldsymbol{x}^2[i]}, \tag{9}$$

$$g(\boldsymbol{x}) = \sum_{i=1}^{n} \log(1 + |\boldsymbol{x}[i]|) - \frac{|\boldsymbol{x}[i]|}{2 + 2|\boldsymbol{x}[i]|}. \tag{10}$$

We provide further insights of these functions in Section A.1 of Supplemental material. Table 2 summarizes their applications. Specifically, Eq. (6) is known as quasinorm, and has attracted a lot of attention because it has resulted in theoretical improvements for matrix completion and compressive sensing [22, 49]. The analysis on the quasinorms is valid with and without the constant $\epsilon$. We prefer to add $\epsilon$ in order to formally satisfy the Lipschitz continuity in Definition 1. Eqs. (7) and (8) enhance the convex $\ell_1$-norm regularizer, and they have significantly improved image denoising [47]. Eq. (9) has been used as the loss function to improve support vector classification [50].

Table 2: List of invex functions studied in this work.

| Reference | Invex function | Application |
|---|---|---|
| [22, 33, 51, 52] | Eq. (6) | Matrix completion |
| [20, 37, 53, 48] | Eq. (7) | Enhancing compressive sensing |
| [47, 54, 55] | Eq. (8) | Image denoising |
| [50] | Eq. (9) | Support vector classification |
| Proposed | Eq. (10) | Compressive sensing |

We propose the last function in Eq. (10) by the subtraction between Eq. (7) and Eq. (8). This design is motivated by the optimization framework in [37] where the regularization term is assumed to be the subtraction of two convex functions (see GIST in Table 1). This has been found to be highly successful in imaging applications (see the survey [48]). But until now there is no evidence that this subtraction produces another convex function (if exists) potentially useful in imaging applications. Therefore, we propose this example to show that at least this is possible in the invex case.

Additionally, we present another way of constructing an invex function in the following lemma. It establishes that an invex function $f : \mathbb{R}^m \to \mathbb{R}$ composed with an affine mapping $\boldsymbol{Hx} - \boldsymbol{b}$ for $\boldsymbol{H} \in \mathbb{R}^{m \times n}$, $\boldsymbol{x} \in \mathbb{R}^n$ and $\boldsymbol{b} \in \mathbb{R}^m$, is also invex if $\boldsymbol{H}$ is full row-rank. This condition on $\boldsymbol{H}$ is a mild assumption, because we show in Section 4 imaging application examples that satisfy this criterium.

**Lemma 2 (Affine Invex Construction).** Let $f : \mathbb{R}^m \to \mathbb{R}$ be a continuously differentiable invex function, $\boldsymbol{H} \in \mathbb{R}^{m \times n}$ have full row rank, and $\boldsymbol{b} \in \mathbb{R}^m$ be a vector. Then the function $h(\boldsymbol{x}) = f(\boldsymbol{Hx} - \boldsymbol{b})$ is invex.

Similar to Lemma 1, it is proved by showing that the composed function satisfies Statement 2 of Theorem 2 (see Appendix B of supplementary material). Eq. (9) is an example of such an invex construction that satisfies the continuously differentiable assumption in Lemma 2, easily verified its proof in Appendix B. A practical implication of Lemma 2 for imaging applications appears when we want to solve linear system of equations (e.g. [50]). We demonstrate an application of this kind of invex construction in Section 4.2.2 to improve a widely used image restoration framework.

## 4 Invex Imaging Examples, Algorithms and Convergence Analysis

In this section, we demonstrate the use of invex regularizers to improve some advanced imaging methodologies. To benefit both practitioners and theory development, we present practical invex imaging algorithms and prove their convergence guarantees to global optima which was only possible for convex functions.

### 4.1 Image Denoising

Image denoising plays a critical role in modern signal processing systems since images are inevitably contaminated by noise during acquisition, compression, and transmission, leading to distortion and loss of image information [56]. Plenty of denoising methods exist, originating from a wide range of disciplines such as probability theory, statistics, partial differential equations, linear and nonlinear filtering, spectral and multiresolution analysis, also classical machine learning and deep learning [57, 58, 56]. All these methods rely on some explicit or implicit assumptions about the true (noise-free) signal in order to separate it properly from the random noise.

One of the most successful assumptions is that a signal can be well approximated by a linear combination of few basis elements in a transform domain [59, 60]. Under this assumption, a denoising method can be implemented as a two-step procedure: i) to obtain high-magnitude transform coefficients that convey mostly the true-signal energy, ii) to discard the transform coefficients which are mainly due to noise. Typical choices for the first step are the wavelet, cosine transforms, and principal component analysis (PCA) [59, 60, 61]. The second step is seen as a filtering procedure that is formally modelled as a proximal optimization problem [62]

$$\text{Prox}_g(\boldsymbol{u}) = \underset{\boldsymbol{x}\in\mathbb{R}^n}{\arg\min}\left(g(\boldsymbol{x}) + \frac{1}{2}\|\boldsymbol{x} - \boldsymbol{u}\|_2^2\right), \tag{11}$$

where $g(\boldsymbol{x})$ acts as a regularization term, and $\boldsymbol{u}$ represents the noisy transform coefficients. In fact, the usefulness of Eq. (11) is not just limited to denoising, but other imaging problems like computer tomography [63], optical imaging [64], biomedical and spectral imaging [65]. In general, global optima guarantees in Eq. (11) is restricted to convex $g(\boldsymbol{x})$, e.g., $\ell_1$-norm.

We improve this important proximal operator by incorporating invex regularizers. Specifically, using those invex functions $g(\boldsymbol{x})$ as listed in Table 2, global minimization is achieved in Eq. (11). The result is presented in the following theorem:

**Theorem 3** (**Invex Proximal**). Consider the optimization problem in Eq. (11) for all functions in Table 2. Then the following holds:

1. The function $h(\boldsymbol{x}) = g(\boldsymbol{x}) + \frac{1}{2}\|\boldsymbol{x} - \boldsymbol{u}\|_2^2$ is convex (therefore invex).

2. The resolvent operator of the proximal is $(\mathbf{I} + \partial g)^{-1}$ and it is treated as a singleton because it always maps to a global optimizer.

It is classically known that the sum of two invex functions is not necessarily invex in general [46]. Therefore, presenting examples like above, where the sum of $f(\boldsymbol{x})$ and $g(\boldsymbol{x})$ is invex, is important to both invexity and imaging communities. We present the proof of Theorem 3 and provide the solution to Eq. (11) for each function in Table 2 in Appendix C of supplementary material.

### 4.2  Image Compressive Sensing

Image *compressive sensing* has been extensively exploited in areas such as microscopy, holography, optical imaging and spectroscopy [66, 67, 68]. It is an inverse problem that aims at recovering an image $\boldsymbol{f} \in \mathbb{R}^n$ from its measurement data vector $\boldsymbol{b} = \boldsymbol{\Phi}\boldsymbol{f}$, where $\boldsymbol{\Phi} \in \mathbb{R}^{m \times n}$ is the image acquisition matrix ($m < n$). Since $m < n$, compressive sensing assumes $\boldsymbol{f}$ has a $k$-sparse representation $\boldsymbol{x} \in \mathbb{R}^n$ ($k \ll n$ non-zero elements) in a basis $\boldsymbol{\Psi} \in \mathbb{R}^{n \times n}$, that is $\boldsymbol{f} = \boldsymbol{\Psi}\boldsymbol{x}$, in order to ensure uniqueness under some conditions. Examples of this sparse basis $\boldsymbol{\Psi}$ in imaging are the Wavelet (also Haar Wavelet) transform, cosine and Fourier representations [69]. Hence, one can work with the abstract model $\boldsymbol{b} = \boldsymbol{\Phi}\boldsymbol{\Psi}\boldsymbol{x} = \boldsymbol{H}\boldsymbol{x}$, where $\boldsymbol{H}$ encapsulates the product between $\boldsymbol{\Phi}$, and $\boldsymbol{\Psi}$, with $\ell_2$-normalized columns [66, 70]. Under this setup, compressive sensing enables to recover $\boldsymbol{x}$ using much lesser samples than what are predicted by the Nyquist criterion [70]. The task formulation is

$$\underset{\boldsymbol{x}\in\mathbb{R}^n}{\text{minimize}}\ f(\boldsymbol{x}) + \lambda g(\boldsymbol{x}) = \frac{1}{2}\|\boldsymbol{H}\boldsymbol{x} - \boldsymbol{b}\|_2^2 + \lambda g(\boldsymbol{x}), \tag{12}$$

where $\lambda \in (0, 1]$ is a typical choice in practice. When the regularizer $g(\boldsymbol{x})$ takes the convex form of $\ell_1$-norm, and when the sampling matrix $\boldsymbol{H}$ satisfies the *restricted isometry property* (RIP) for any $k$-sparse vector $\boldsymbol{x} \in \mathbb{R}^n$, i.e., $(1 - \delta_{2k})\|\boldsymbol{x}\|_2^2 \leq \|\boldsymbol{H}\boldsymbol{x}\|_2^2 \leq (1 + \delta_{2k})\|\boldsymbol{x}\|_2^2$ for $\delta_{2k} < \frac{1}{3}$ [69, Theorem 6.9], it has been proved that $\boldsymbol{x}$ can be exactly recovered by solving Eq. (12) [71].

We are interested in invex regularizers. It has been proved that, when $g(\boldsymbol{x})$ takes the particular invex form in Eq. (6), $\boldsymbol{x}$ can be exactly recovered by solving Eq. (12) [49]. Below we further generalize this result to all the invex functions as listed in Table 2. The generalized result is presented in Theorem 4.

**Theorem 4** (**Invex Image Compressive Sensing**). Assume $\boldsymbol{H}\boldsymbol{x} = \boldsymbol{b}$, where $\boldsymbol{x} \in \mathbb{R}^n$ is $k$-sparse, the matrix $\boldsymbol{H} \in \mathbb{R}^{m \times n}$ ($m < n$) with $\ell_2$-normalized columns that satisfies the RIP condition for any $k$-sparse vector, and $\boldsymbol{b} \in \mathbb{R}^m$ is a noiseless measurement vector. If $g(\boldsymbol{x})$ in Eq. (12) takes the form of the functions in Table 2, then the following holds:

1. The objective function $\frac{1}{2}\|\boldsymbol{H}\boldsymbol{x} - \boldsymbol{b}\|_2^2 + \lambda g(\boldsymbol{x})$ is invex.

2. $\boldsymbol{x}$ can be exactly recovered by solving Eq. (12) i.e. only global optimizers exist. When $g(\boldsymbol{x})$ takes the form of Eq. (9), extra mild conditions on $\boldsymbol{x}$ are needed.

We clarify that if $\boldsymbol{H}$ satisfies the mentioned RIP, then each sub-matrix with $k$-columns of $\boldsymbol{H}$, selected according to indices of the nonzero elements of the $k$-sparse signal is a full row-rank matrix. This result is important to invex community, because it supports the validity of Lemma 2 to build invex functions with affine mappings. Additionally, we present another proved form of function sum that can result in an invex function, i.e., the sum of $g(\boldsymbol{x})$ and the $\ell_2$-norm composed with the affine mapping $\boldsymbol{H}\boldsymbol{x} - \boldsymbol{b}$. The complete proof is provided in Appendix E of supplementary material.

Next, we present different algorithms to solve Eq. (12) using invex $g(\boldsymbol{x})$ as in Table 2. We select a few of the most important and successful image reconstruction techniques to start from, and develop their invex extensions. Taking advantage of the invex property, we prove convergence to global minimizers for each extended algorithm, which is unexplored up to date.

---

**Algorithm 1** Accelerated Proximal Gradient

1: **input**: Tolerance constant $\epsilon \in (0, 1)$, initial point $\boldsymbol{x}^{(0)}$, and number of iterations $T$.
2: **initialize**: $\boldsymbol{x}^{(1)} = \boldsymbol{x}^{(0)} = \boldsymbol{z}^{(0)}, r_1 = 1, r_0 = 0, \alpha_1, \alpha_2 < \frac{1}{L}$, and $\lambda \in (0, 1]$
3: **for** $t = 1$ to $T$ **do**
4: $\quad \boldsymbol{y}^{(t)} = \boldsymbol{x}^{(t)} + \frac{r_{t-1}}{r_t}(\boldsymbol{z}^{(t)} - \boldsymbol{x}^{(t)}) + \frac{r_{t-1}-1}{r_t}(\boldsymbol{x}^{(t)} - \boldsymbol{x}^{(t-1)})$
5: $\quad \boldsymbol{z}^{(t+1)} = \text{prox}_{\alpha_2 \lambda g}(\boldsymbol{y}^{(t)} - \alpha_2 \nabla f(\boldsymbol{y}^{(t)}))$
6: $\quad \boldsymbol{v}^{(t+1)} = \text{prox}_{\alpha_1 \lambda g}(\boldsymbol{x}^{(t)} - \alpha_1 \nabla f(\boldsymbol{x}^{(t)}))$
7: $\quad r_{t+1} = \frac{\sqrt{4(r_t)^2+1}+1}{2}$
8: $\quad \boldsymbol{x}^{(t+1)} = \begin{cases} \boldsymbol{z}^{(t+1)}, & \text{if } f(\boldsymbol{z}^{(t+1)}) + \lambda g(\boldsymbol{z}^{(t+1)}) \leq f(\boldsymbol{v}^{(t+1)}) + \lambda g(\boldsymbol{v}^{(t+1)}) \\ \boldsymbol{v}^{(t+1)}, & \text{otherwise} \end{cases}$
9: **end for**
10: **return**: $\boldsymbol{x}^{(T)}$

---

### 4.2.1 Accelerated Proximal Gradient Algorithm

The accelerated proximal gradient (APG) method [72] has been shown to be effective solving Eq. (12), achieving better imaging quality in less iterations than its predecessors [13, 36, 37, 38, 73], and been frequently used by recent imaging works [55, 74, 75, 76]. Its convergence to global optima is only guaranteed for convex loss [72]. For non-convex cases, convergence to a critical point has been stated [72]. Its pseudo-code for solving Eq. (12) is provided in Algorithm 1.

Taking advantage that the loss function $f(\boldsymbol{x}) + \lambda g(\boldsymbol{x})$ in Eq. (12) is invex, and the uniqueness result in Theorem 3, we formally extend APG in the following lemma stating that the sequence $\{\boldsymbol{x}^{(t+1)}\}$ generated by Algorithm 1 converges to a global minimizer of Eq. (12).

**Lemma 3 (Invex APG).** Under the setup of Theorem 4 and using $L = \sigma_1\left(\boldsymbol{H}^T\boldsymbol{H}\right)$ (maximum singular value), the sequence $\{\boldsymbol{x}^{(t)}\}_{t=0}^{T-1}$ generated by Algorithm 1 converges to a global minimizer.

To prove Lemma 3, we apply the Statement 2 of Theorem 2 to Eq. (12) and the unicity of the proximal operators for functions in Table 2. The proof is provided in Appendix F of supplementary material.

### 4.2.2 Plug-and-play with Deep Denoiser Prior

Plug-and-play (PnP) is a powerful framework for regularizing imaging inverse problems [65] and has gained popularity in a range of applications in the context of imaging inverse problems [29, 65, 77, 78, 79]. It replaces the proximal operator in an iterative algorithm with an image denoiser, which does not necessarily have a corresponding regularization objective. This implies that the effectiveness of PnP goes beyond standard proximal algorithms such as primal-dual splitting [80, 81, 82]. It has guarantees to a fixed point only when convex objective functions are employed [81].

To apply the PnP framework, we modify Algorithm 1 by replacing the proximal operator (Line 6 in its pseudo-code) with a neural network based denoiser Noise2Void [58], resulting in

$$\boldsymbol{v}^{(t+1)} = \text{Noise2Void}\left(\boldsymbol{x}^{(t)} - \alpha_1 \nabla f\left(\boldsymbol{x}^{(t)}\right)\right). \tag{13}$$

The complete pseudo-code is presented in Algorithm 3 of Appendix G in supplemental material. We remark that in Algorithm 3, Line 5 of the Algorithm 1 is retained to allows the comparison between regularizers (invex and convex). More specifically, Line 5 computes the proximal step, while Line 6 relies on a neural network for the same purpose (13). This offers an avenue for simultaneously exploiting both the model-based and data-driven approaches. The output of Algorithm 3 is a close estimation to the solution of Eq. (12) [81]. The benefit of using this denoiser is that it does not require clean target images in order to be trained. We present the following convergence result for this modified algorithm under the assumption of $f(\boldsymbol{x})$ in Eq. (12) being invex which is a generalization of [81] (restricted to convex functions only).

**Lemma 4** (**Invex Plug-and-play**). Assume $f(\boldsymbol{x})$ in Eq. (12) is invex with Lipschitz continuous gradient, and a denoiser $d : \mathbb{R}^n \to \mathbb{R}$. Under the setup of Theorem 4 and some mild conditions on $d$, the sequence $\left\{\boldsymbol{x}^{(t)}\right\}_{t=0}^T$ generated by Algorithm 1 satisfies

$$\frac{1}{T} \sum_{t=1}^{T} \left\| \boldsymbol{x}^{(t)} - d\left(\boldsymbol{x}^{(t)} - \alpha_1 \nabla f\left(\boldsymbol{x}^{(t)}\right)\right) \right\|_2^2 \leq \frac{2}{T} \left(\frac{1+\kappa}{1-\kappa}\right) \left\| \boldsymbol{x}^{(0)} - \boldsymbol{x}^* \right\|_2^2, \qquad (14)$$

for any $\boldsymbol{x}^* = d(\boldsymbol{x}^* - \alpha_1 \nabla f(\boldsymbol{x}^*))$ (fixed point) and for some $\kappa \in (0, 1)$.

Eq. (14) guarantees that $\left\{\boldsymbol{x}^{(t)}\right\}_{t=0}^T$ is arbitrarily close to the set of fixed points of $d(\cdot)$, which is considered a close estimation to the solution of Eq. (12) [81]. Its proof is provided in Appendix G of supplementary material. Eq. (9) is an example satisfying assumption required in Lemma 4.

### 4.2.3 Unrolling

The *unrolling* or *unfolding* framework is another imaging strategy for solving Eq. (12). It offers a systematic connection between iterative algorithms used in signal processing and the neural networks [31, 32, 83]. Unrolled neural networks become popular due to their potential in developing efficient and high-performing network architectures from reasonably sized training sets [84, 85]. A folded version of the proximal gradient algorithm is presented in Algorithm 2. Particularly, existing works [86, 87] have shown that the efficiency of Algorithm 2 can be improved by simulating a recurrent neural network so that its layers mimic the iterations in Line 4 of Algorithm 2. Specifically, each $\boldsymbol{x}^{(t+1)}$ constitutes one linear operation which models a layer of the network, followed by a proximal operation that models the activation function. Thus, one forms a deep network by mapping each iteration to a network layer and stacking the layers together to learn $\boldsymbol{H}, \alpha_t$, and $\boldsymbol{x}^{(t)}$ for all $t$ which is equivalent to executing an iteration of Algorithm 2 multiple times. Their study was conducted only for $g(\boldsymbol{x})$ in the form of $\ell_1$-norm.

---

**Algorithm 2** Folded Proximal Gradient Algorithm

---

1: **input**: initial point $\boldsymbol{x}^{(0)}$, number of iterations $T$
2: **initialize**: $\alpha_t < \frac{2}{L+2}$, and $\lambda \in (0, 1]$
3: **for** $t = 0$ to $T$ **do**
4: $\quad \boldsymbol{x}^{(t+1)} = \text{prox}_{\alpha_t \lambda g}(\boldsymbol{x}^{(t)} - \alpha_t \boldsymbol{H}^T(\boldsymbol{H}\boldsymbol{x}^{(t)} - \boldsymbol{b}))$
5: **end for**
6: **return:** $\boldsymbol{x}^{(T)}$

---

Convergence guarantees to global optima for Algorithm 2 has been established in [13], but it is restricted to convex objective functions. Therefore, due to the success and importance of unrolling we aim to extend the global optima guarantees of Algorithm 2 to invex objectives, and present the results in the following lemma:

**Lemma 5** (**Invex Unrolling**). Under the setup of Theorem 4 and using $L = \sigma_1\left(\boldsymbol{H}^T \boldsymbol{H}\right)$ (maximum singular value) and $\alpha_t < \frac{2}{L+2}$, the sequence $\left\{\boldsymbol{x}^{(t)}\right\}_{t=0}^{T-1}$ generated by Algorithm 2 converges to a global minimizer.

The key to proving Lemma 5 relies on the uniqueness result of the proximal operator for functions in Table 2 as stated in Theorem 3. The proof is presented in Appendix H of supplementary material. Such results confirm that the invex unrolled network of Algorithm 2, which uses the proximal operators of invex mappings as the activation functions, can reach the optimal solution during training.

# 5 Experiments and Results

A number of datasets have been merged to formulate one unique dataset for our training and evaluation purposes. These are DIV2K super-resolution [88], the McMaster [89], Kodak [90], Berkeley Segmentation (BSDS 500) [91], Tampere Images (TID2013) [92] and the Color BSD68 [93] datasets. We conduct various experiments to study the performance of those invex regularizers as listed in Table 2 in non-ideal conditions. We compare them against the state-of-the-art methods originally developed for convex regularizers ($\ell_1$-norm) ensuring global optima. When neural network training is involved, we take a total of 900 images which are randomly divided into a training set of 800 images, a validation set of 55 images, and a test set of 45 images. For all the experiments, the images are scaled into the range $[0, 1]$. For the invex regularizer in Eq. (6), we vary the value of $p$.

## 5.1 Image Compressive Sensing Experiments

We assess signal reconstruction, in these experiments, by averaging the peak-signal-to-noise-ratio (PSNR) in dB over the testing image set. We consider additive white Gaussian noise in the measurements data vector with three different levels of SNR (Signal-to-Noise Ratio) = 20, 30, and $\infty$ (noiseless case). For Algorithm 1 and its plug-and-play variant, the parameters $\lambda, \alpha_1$, and $\alpha_2$ were chosen to be the best for each analyzed function determined by cross validation, and the initial point $\boldsymbol{x}^{(0)}$ was the blurred image $\boldsymbol{b}$. The results are summarized in Table 3, where the best and least efficient among invex functions is highlighted in boldface and underscore, respectively. Additional results are reported in Appendix I of supplemental material for each experiment, using the structural similarity index measure to assess imaging quality.

**Experiment 1** studies the effect of different invex regularizers, the Smoothly Clipped Absolute Deviation (SCAD) [94], and the Minimax Concave Penalty (MCP) [24], under Algorithm 1. A deconvolution problem is studied to formulate Eq. (12) which is an important problem in signal processing due to imperfect artefacts in physical setups such as mismatch, calibration errors, and loss of contrast [95]. To compare, the used state-of-the-art methods that employ convex regularization are the Total Variation Minimization by Augmented Lagrangian (TVAL3) [96], and the fast iterative shrinkage-thresholding algorithm (FISTA) [13] which ensures global optima. Further, to comparing with convolutional neural networks methodologies, the non-iterative reconstruction methodology ReconNet [97] is used. To model this problem, all pixels of the testing set are fixed to $256 \times 256$ pixels. The images went through a Gaussian blur of size $9 \times 9$ and standard deviation $4$, followed by an additive zero-mean white Gaussian noise. The sensing matrix $\boldsymbol{H}$ is built as $\boldsymbol{H} = \boldsymbol{\Phi}\boldsymbol{\Psi}$ (for all methods except ReconNet), where $\boldsymbol{\Phi}$ represents the blur operator over the images and $\boldsymbol{\Psi}$ is the inverse of a three stage Haar wavelet transform. This experiment is extremely ill-conditioned, where the condition number of $\boldsymbol{H}^T\boldsymbol{H}$ is significantly higher than 1. This means that in practice the RIP condition is not guaranteed. To achieve a fair comparison, the number of iterations was fixed for all functions as $T = 800$. The deconvolution problem follows a compressive sensing setup because the Gaussian filter remove high frequency information of the input image.

In the case of ReconNet, we follow existing setting in [97]. For the learning of ReconNet, we extract patches of size $33 \times 33$ from the noisy blurred training image set, and we train it using the Adam optimization algorithm and a learning rate $5 \times 10^{-4}$ for 512 epochs with a batch size of 128.

**Experiment 2** studies the invex regularizers under the plug-and-play modification of Algorithm 1 as described in Section 4.2.2 [2] [58]. The same deconvolution problem as in Experiment 1 is used. The interesting aspect of this scenario is that Algorithm 1 has a proximal step in Line 5 that allows to compare between regularizers (invex and convex) while using neural networks in Line 6 (see Algorithm 3 in Appendix G of Supplemental material). Noise2Void is trained by randomly extracting patches of size $64 \times 64$ pixels from the training images where zero-mean white Gaussian noise was added for $SNR = 20, 30$dB. Data augmentation on the training dataset is used, by rotating each image three times by 90 and also added all mirrored versions. The learning rate is fixed as 0.0004.

**Experiment 3** compares the invex regularizers but under the unrolling framework as described in Section 4.2.3. The gold standard convex regularizations to compare with are the learned iterative shrinkage and thresholding algorithm (LISTA) [87], and the Interpretable optimization-inspired deep network (ISTA-Net)[98]. Also, to comparing with convolutional neural networks methodologies,

---

[2]We used Noise2Void implementation at `https://github.com/juglab/n2v`

Table 3: Performance comparison, in terms of PSNR (dB), where the best and least efficient among invex functions is highlighted in boldface and underscore, respectively.

| (Experiment 1) Algorithm 1, $p = 0.5$ for Eq. (6). | | | | | | FISTA [13] | ReconNet [97] | TVAL3 [96] | SCAD [94] | MCP [24] |
|---|---|---|---|---|---|---|---|---|---|---|
| SNR | Eq. (6) | Eq. (7) | Eq. (8) | Eq. (9) | Eq. (10) | $\ell_1$-norm | | | | |
| ∞ | **33.40** | 31.25 | 31.93 | 30.00 | 32.65 | 29.97 | 27.01 | 28.77 | 30.55 | 31.30 |
| 20dB | **24.60** | 22.83 | 23.39 | 22.00 | 23.98 | 21.80 | 19.99 | 20.49 | 22.60 | 23.01 |
| 30dB | **27.61** | 26.56 | 26.90 | 26.00 | 27.25 | 24.91 | 22.01 | 23.99 | 26.10 | 26.77 |

| (Experiment 2) Algorithm 3, $p = 0.8$ for Eq. (6). | | | | | | |
|---|---|---|---|---|---|---|
| SNR | Eq. (6) | Eq. (7) | Eq. (8) | Eq. (9) | Eq. (10) | $\ell_1$-norm |
| ∞ | **34.51** | 32.37 | 33.06 | 31.40 | 33.76 | 31.10 |
| 20dB | **25.55** | 23.92 | 24.44 | 23.00 | 24.98 | 22.95 |
| 30dB | **28.30** | 26.87 | 27.33 | 26.05 | 27.80 | 26.00 |

| (Denoising experiment) Algorithm 4, $p = 0.5$ for Eq. (6) | | | | BM3D [59] | Noise2Void [58] |
|---|---|---|---|---|---|
| Metric | Eq. (6) | Eq. (8) | Eq. (10) | $\ell_1$-norm | |
| SNR (dB) | **49.40** | 43.85 | 46.46 | 41.52 | 39.43 |
| SSIM | **0.886** | 0.872 | 0.876 | 0.869 | 0.853 |

| (Experiment 3) Algorithm 2 - unfolded LISTA. $p = 0.85$ for Eq. (6) | | | | | | | | |
|---|---|---|---|---|---|---|---|---|
| SNR | $m/n$ | Eq. (6) | Eq. (7) | Eq. (8) | Eq. (9) | Eq. (10) | $\ell_1$-norm [86] | ReconNet [97] |
| | 0.2 | **31.32** | 29.20 | 29.87 | 28.56 | 30.58 | 27.95 | 26.59 |
| ∞ | 0.4 | **36.10** | 33.50 | 34.34 | 32.75 | 35.20 | 32.01 | 31.86 |
| | 0.6 | **41.27** | 37.81 | 38.90 | 36.09 | 40.05 | 35.82 | 34.42 |
| | 0.2 | **26.00** | 24.45 | 24.94 | 23.97 | 25.01 | 23.52 | 22.00 |
| 20dB | 0.4 | **32.67** | 30.64 | 31.32 | 30.02 | 32.29 | 29.43 | 28.24 |
| | 0.6 | **34.38** | 33.00 | 33.28 | 32.94 | 33.64 | 32.60 | 30.20 |
| | 0.2 | **27.65** | 26.20 | 26.66 | 25.75 | 27.15 | 25.32 | 23.64 |
| 30dB | 0.4 | **34.33** | 31.89 | 32.66 | 31.02 | 33.47 | 30.46 | 29.88 |
| | 0.6 | **37.03** | 34.84 | 35.54 | 34.17 | 36.27 | 33.53 | 31.71 |

| (Experiment 3) Algorithm 2 - unfolded ISTA-Net. $p = 0.85$ for Eq. (6) | | | | | | | |
|---|---|---|---|---|---|---|---|
| SNR | $m/n$ | Eq. (6) | Eq. (7) | Eq. (8) | Eq. (9) | Eq. (10) | $\ell_1$-norm [98] |
| | 0.2 | **32.50** | 30.15 | 30.89 | 29.04 | 31.67 | 28.77 |
| ∞ | 0.4 | **38.33** | 35.72 | 36.55 | 34.92 | 37.41 | 34.17 |
| | 0.6 | **43.61** | 40.07 | 41.18 | 39.02 | 42.36 | 38.02 |
| | 0.2 | **28.29** | 26.22 | 26.87 | 25.60 | 27.56 | 25.01 |
| 20dB | 0.4 | **33.96** | 32.11 | 32.71 | 31.55 | 33.32 | 31.00 |
| | 0.6 | **35.77** | 34.68 | 35.03 | 34.33 | 35.39 | 33.99 |
| | 0.2 | **29.34** | 28.30 | 28.63 | 27.97 | 28.98 | 27.65 |
| 30dB | 0.4 | **35.41** | 33.33 | 33.99 | 32.69 | 34.68 | 32.08 |
| | 0.6 | **38.95** | 36.25 | 37.10 | 35.43 | 38.00 | 34.65 |

the non-iterative reconstruction methodology ReconNet [97] is used. We follow the existing setting for LISTA in [86][3], and for ISTA-Net in [87]. For the training stage we extract 10000 patches $b \in \mathbb{R}^{16 \times 16}$ at random positions of each image, with all means removed. We then learn a dictionary $D \in \mathbb{R}^{256 \times 512}$ from the extracted patches, using the same strategy as in [86]. Gaussian i.i.d sensing matrices $\Phi \in \mathbb{R}^{m \times 256}$ are created from the standard Gaussian distribution, $\Phi[i, j] \sim \mathcal{N}(0, 1/m)$ and then normalize its columns to have the unit $\ell_2$-norm, where $m$ is selected such that $\frac{m}{256} = 0.2, 0.4, 0.6$. The matrix $H$ is built as $H = \Phi\Psi$ with $T = 16$ (number of layers). We follow the same two-step strategy in [86] to train a recurrent neural network. First, perform a layer-wise pre-training solving Eq. (12) for each extracted patch $b$ by fixing $H = \Psi$. Second, append a learnable fully-connected layer at the end of the network structure, initialized by $\Psi$. Then, perform an end-to-end training solving Eq. (12) where $H$ in this case is learnt by updating the initial matrix $\Psi$. For each testing image, we divide it into non-overlapping $16 \times 16$ patches. When $g(x)$ is the the $\ell_1$-norm, we recover [86].

In the case of ISTA-Net, and ReconNet, for their learning stage we extract patches from the training image set of size $33 \times 33$. Gaussian i.i.d sensing matrices $\Phi \in \mathbb{R}^{m \times 1089}$ are created with $\ell_2$-normalized columns as for LISTA, where $m$ is selected such that $\frac{m}{1089} = 0.2, 0.4, 0.6$. The optimizer employed was Adam algorithm and a learning rate $1 \times 10^{-4}$ for 200 and 512 epochs for ISTA-Net and ReconNet respectively, with a batch size of 64 for both networks. For ISTA-Net $T = 16$ (number of unrolled iterations). We recall that when $g(x)$ is the the $\ell_1$-norm in ISTA-Net, we recover [98].

## 5.2 Image Denoising Experiment

Two image datasets, which we merge (80 images in total), are used for this experiment comes from a neutron image formation phenomenon[4]. These type of images contain the neutron attenuation properties of the object which helps analyze material structure. Performance is assessed by averaging along all the images the experimental SNR in dB given by $SNR = 20 \log\left(\frac{\|z\|_2}{\|\hat{z} - z\|_2}\right)$, where $z$ and $\hat{z}$ stand for the noisy and the denoised image, respectively, and the structural similarity index measure (SSIM) computed between $z$ and $\hat{z}$. Taking advantage of results observed from previous experiments, we compare the top three regularizers in Eqs. (6), (8), and (10) with two state-of-the-art denoising techniques including the block-matching and 3-D filtering (BM3D) [59] using $\ell_1$-norm regularizer and the deep learning technique Noise2Void (trained as in Experiment 2) [58]. We follow the two-step denoising procedure described in Section 4.1. In the first step, the transform domain is built using

---

[3]We used the implementation from [86] at `https://github.com/VITA-Group/LISTA-CPSS`

[4]Acquired with the ISIS Neutron and Moun Source system at Harwell Science and Innovation Campus.

PCA as in [61]. To build this transform we extract patches of $16 \times 16$ from the noisy image that are then used to adaptively construct a tight frame (nearly orthogonal matrix) tailored to the given noisy data [5]. Results are summarized in Table 3. We report examples of denoised images obtained by Eqs. (6), (8), (10), BM3D, and Noise2Void are illustrated in Appendix J of supplementary material, along with the algorithm used for the invex regularizers to denoise these images.

## 6    Discussion, Limitations and Conclusion

Application advancement of invex theory has paused for decades due to the lack of practical examples, which has caused a significantly reduced interest in invexity research. To address this issue, we present for the first time a list of invex regularizers for image reconstruction applications, and formulate corresponding optimization problems. Particularly, for image compressive sensing, we improve three advanced imaging techniques using the listed functions in Table 2 as invex regularizers. We present their solution algorithms and develop theoretical guarantees on their convergence to global minimum. We also conducted various image compressive sensing and denoising experiments to demonstrate the effectiveness of invex regularizers under practical scenarios that are non-ideal with noisy data observed and RIP condition not guaranteed. Significant benefit of using invex regularizers have been proved from both theoretical and empirical aspects. In fact, Table 3 and theoretical results in Section 4 revive the potential of exploring invex theory in practical applications.

The numerical results presented in Table 3 confirm performance improvement by using invex regularizers over the $\ell_1$-norm-based methods (e.g FISTA, TVAL3) in unexplored scenarios. These tables and theoretical results in Section 4 revive the potential of exploring invex theory in practical applications. The best result is obtained with Eq. (6), and Eq. (9) is the least efficient. The intuition behind the superiority of Eq. (6) comes from the possibility of adjusting the value of $p$ in data-dependent manner [49]. This means that when the images are strictly sparse, and the noise is relatively low, a small value of $p$ should be used. Conversely, when images are non-strictly sparse and/or the noise is relatively high, a larger value of $p$ tend to yield better performance (which seems to be the case for the selected image datasets). We believe that the remaining invex, SCAD, and MCP regularizers have a lesser performance than Eq. (6) as they do not have the flexibility of adjustment to the sparsity of the data. In fact, Eq. (9) shows the poorest performance because in the proof of Theorem 4, we theoretically guarantee that Eq. (9) cannot sparsify all images. Therefore, this analysis leads to the conclusion that the invex function Eq. (6) offers the best performance for the metrics concerns and the imaging problems studied here.

Although, we have presented theoretical results with global optima using invexity for some of most important and successful image reconstruction techniques, we highlight several limitations of our analysis. Specifically, we focused on reconstructed the image of interest in an ideal scenario, that is, without the present of noise (Theorem 4). Additionally, we have limited our numerical results to tasks like denoising, and deconvolution. And, the convergence guarantees for the plug-and-play result only ensures a close estimate of the solution (Lemma 4). Therefore, we see there are a number of future directions this research can be taken further improving the results even further. One aspect is to explore avenues for improving convergence guarantees to global optima the plug-and-play framework. Another direction is the study of inclusion of noise in the analysis of imaging applications, which may be an enabler to improve downstream tasks like invex robust image reconstruction. Finally, we feel that the application domains for invex functions can go well beyond denoising, and deconvolution imaging problems, especially around deep learning research, which can improved a number of downstream applications.

### Broader Impact

We believe that the presented mathematical and empirical analysis over the studied regularizers has the potential to unlock the benefits of invexity for further applications in signal and image processing. This may be an enabler to improve downstream tasks like deep learning for imaging, and to provide more robust image reconstruction algorithms.

### Acknowledgments

This work was partially supported by the Facilities Funding from Science and Technology Facilities Council (STFC) of UKRI, and Wave 1 of the UKRI Strategic Priorities Fund under the EPSRC grant EP/T001569/1, particularly the "AI for Science" theme within that grant, by the Alan Turing Institute.

---

[5]We used implementation at `https://www.math.hkust.edu.hk/~jfcai/`.

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
