# Supplementary Material
# Improved Imaging by Invex Regularizers
# with Global Optima Guarantees

**Samuel Pinilla[1,2], Tingting Mu[3], Neil Bourne[2], Jeyan Thiyagalingam[1]**

[1*] Scientific Computing Department, Science and Technology Facilities Council, Harwell, UK
[2] University of Manchester at Harwell, UK
[3] Computer Science, University of Manchester, UK
{samuel.pinilla,t.jeyan}@stfc.ac.uk
{tingting.mu,neil.bourne}@manchester.ac.uk

## 1 Proof of Lemma 1

In this proof we seek to guarantee that the list of functions in Table 2 are invex. We point out that, since the regularizers in Table 2 is the sum of a scalar function applied to each entry of a vector, then it is enough to analyze the scalar function to determine the invexity of the regularizer.

**Eq. (6).**

*Proof.* Take $r_\epsilon(w) = (|w| + \epsilon)^p$, $\forall w \in \mathbb{R}$, for $p \in (0,1)$ and $\epsilon \geq (p(1-p))^{\frac{1}{2-p}}$. The need to add the constant $\epsilon$ it is to formally satisfy the Lipschitz continuous condition required to be invex according to Definition 2. Observe that if $w > 0$ then we have that $\partial r_\epsilon(w) = \left\{ \frac{p}{(|w|+\epsilon)^{1-p}} \right\}$, which means that $0 \notin \partial r_\epsilon(w)$. Conversely, if $w < 0$ then $\partial r_\epsilon(w) = \left\{ \frac{-p}{(|w|+\epsilon)^{1-p}} \right\}$, leading to $0 \notin \partial r_\epsilon(w)$. Lets examinate $w^* = 0$. Note that

$$\lim_{w \to 0^+} r'_\epsilon(w) = \lim_{w \to 0^+} \frac{p}{(|w| + \epsilon)^{1-p}} = \frac{p}{\epsilon^{1-p}}, \tag{1}$$

and that

$$\lim_{w \to 0^-} r'_\epsilon(w) = \lim_{w \to 0^-} \frac{-p}{(|w| + \epsilon)^{1-p}} = \frac{-p}{\epsilon^{1-p}}. \tag{2}$$

Additionally, since $r_\epsilon(w)$ is a Lipschitz continuous function, then appealing to Theorem 1 we have that $\partial r_\epsilon(w^* = 0) = \text{conv} \left\{ \frac{-p}{\epsilon^{1-p}}, \frac{p}{\epsilon^{1-p}} \right\} = \left[ \frac{-p}{\epsilon^{1-p}}, \frac{p}{\epsilon^{1-p}} \right]$. This means that $0 \in \partial r_\epsilon(0)$. Further, given the fact that $r_\epsilon(0) \leq r_\epsilon(w)$ for all $w \in \mathbb{R}$, then $w^* = 0$ is a global minimizer of $r_\epsilon$. Therefore, the function $r_\epsilon$ is invex. $\square$

**Eq. (7)**

*Proof.* Take $r(w) = \log(1 + |w|)$. Observe that if $w > 0$ then we have that $\partial r(w) = \left\{ \frac{1}{1+|w|} \right\}$, which means that $0 \notin \partial r(w)$. Conversely, if $w < 0$ then $\partial r(w) = \left\{ \frac{-1}{1+|w|} \right\}$, leading to $0 \notin \partial r(w)$. Lets examinate $w^* = 0$. Note that

$$\lim_{w \to 0^+} r'(w) = \lim_{w \to 0^+} \frac{1}{1 + |w|} = 1, \tag{3}$$

---

[*]Rutherford Appleton Laboratory.

36th Conference on Neural Information Processing Systems (NeurIPS 2022).

and that

$$\lim_{w \to 0^-} r'(w) = \lim_{w \to 0^-} \frac{-1}{1 + |w|} = -1. \tag{4}$$

Additionally, since $r(w)$ is a Lipschitz continuous function, then appealing to Theorem 1 we have that $\partial r(w^* = 0) = \text{conv } \{-1, 1\} = [-1, 1]$. This means that $0 \in \partial r(0)$. Further, given the fact that $r(0) \leq r(w)$ for all $w \in \mathbb{R}$, then $w^* = 0$ is a global minimizer of $r(w)$. Therefore, the function $r(w)$ is invex. $\qquad \square$

**Eq.** (8)

*Proof.* Take $r(w) = \frac{|w|}{2 + 2|w|}$. Observe that if $w > 0$ then we have that $\partial r(w) = \left\{ \frac{1}{2(1+|w|)^2} \right\}$, which means that $0 \notin \partial r(w)$. Conversely, if $w < 0$ then $\partial r(w) = \left\{ \frac{-1}{2(1+|w|)^2} \right\}$, leading to $0 \notin \partial r(w)$. Lets examinate $w^* = 0$. Note that

$$\lim_{w \to 0^+} r'(w) = \lim_{w \to 0^+} \frac{1}{2(1 + |w|)^2} = \frac{1}{2}, \tag{5}$$

and that

$$\lim_{w \to 0^-} r'(w) = \lim_{w \to 0^-} \frac{-1}{2(1 + |w|)^2} = -\frac{1}{2}. \tag{6}$$

Additionally, since $r(w)$ is a Lipschitz continuous function, then appealing to Theorem 1 we have that $\partial r(w^* = 0) = \text{conv } \left\{-\frac{1}{2}, \frac{1}{2}\right\} = \left[-\frac{1}{2}, \frac{1}{2}\right]$. This means that $0 \in \partial r(0)$. Further, given the fact that $r(0) \leq r(w)$ for all $w \in \mathbb{R}$, then $w^* = 0$ is a global minimizer of $r(w)$. Therefore, the function $r(w)$ is invex. $\qquad \square$

**Eq.** (9)

*Proof.* Consider $r(w) = \frac{w^2}{1 + w^2}$. Observe that $\partial r(w) = \left\{ \frac{2w}{(1 + w^2)^2} \right\}$, which means $r(w)$ is continuously differentiable. Then, it is clear that $w = 0$ is the only point that satisfies $0 \in \partial r(0)$. In addition, the value $r(w = 0)$ is the global minimum of $r(w)$. Thus, since the only stationary point of $r(w)$ is a global minimizer, then $r(w)$ is invex. $\qquad \square$

**Eq.** (10)

*Proof.* Take $r(w) = \log(1 + |w|) - \frac{|w|}{2 + 2|w|}$. Observe that if $w > 0$ then we have that $\partial r(w) = \left\{ \frac{1}{2(1+|w|)^2} + \frac{w}{(1+|w|)^2} \right\}$, which means that $0 \notin \partial r(w)$. Conversely, if $w < 0$ then $\partial r(w) = \left\{ \frac{-1}{2(1+|w|)^2} + \frac{w}{(1+|w|)^2} \right\}$, leading to $0 \notin \partial r(w)$. Lets examinate $w^* = 0$. Note that

$$\lim_{w \to 0^+} r'(w) = \lim_{w \to 0^+} \frac{1}{2(1 + |w|)^2} + \frac{w}{(1 + |w|)^2} = \frac{1}{2}, \tag{7}$$

and that

$$\lim_{w \to 0^-} r'(w) = \lim_{w \to 0^-} \frac{-1}{2(1 + |w|)^2} + \frac{w}{(1 + |w|)^2} = -\frac{1}{2}. \tag{8}$$

Additionally, since $r(w)$ is a Lipschitz continuous function, then appealing to Theorem 1 we have that $\partial r(w^* = 0) = \text{conv } \left\{-\frac{1}{2}, \frac{1}{2}\right\} = \left[-\frac{1}{2}, \frac{1}{2}\right]$. This means that $0 \in \partial r(0)$. Further, given the fact that $r(0) \leq r(w)$ for all $w \in \mathbb{R}$, then $w^* = 0$ is a global minimizer of $r(w)$. Therefore, the function $r(w)$ is invex. $\qquad \square$

## 1.1 Additional Discussion on Invex Regularizers

To address sub-optimal limitations of convex regularizers, non-convex mappings have been proposed. For instance, the Smoothly Clipped Absolute Deviation (SCAD) [1], and Minimax Concave Penalty (MCP) [2]. However, a recent survey in imaging [3], which compared the performance of several regularizers including SCAD and MCP for a number of imaging, concludes that Eq. (6) shows higher performance than SCAD and MCP because the value of $p$ can be adjusted in data-dependent manner. This means that when the images are strictly sparse, and the noise is relatively low, a small value of $p$ should be used. Conversely, when images are non-strictly sparse and/or the noise is relatively high, a larger value of $p$ tend to yield better performance. Furthermore, in the context of invexity, we highlight that SCAD and MCP are non-invex regularizer because they reach a maximum value, which makes the first derivative zero in non-minimizer values leading to its non-invexity (see Theorem 1).

On the other hand, in the case of minimax-concave-type of regularizers, we present a new function in our manuscript (Eq. (10)). From Eq. (10) it is clear we are subtracting $g_1(\boldsymbol{x}) = \sum_{i=1}^{n} \log(1 + |\boldsymbol{x}[i]|)$, and $g_2(\boldsymbol{x}) = \sum_{i=1}^{n} \frac{|\boldsymbol{x}[i]|}{2 + 2|\boldsymbol{x}[i]|}$ (selected due to results in [4]). We propose to study regularizer in Eq. (10), that is $g_1 - g_2$, for three reasons. First, because $g_1$, $g_2$, and $g_1 - g_2$ are invex, as stated in Lemma 1, and all of them can achieve global optima for the scenarios studied in the paper. Second, to the best of our knowledge, there is no evidence that subtracting two convex penalties (current proposal in the minimax-concave literature) produces another convex regularizer (if exists). Therefore, we present Eq. (10) to show that at least this is possible in the invex case, as stated in Section 3.

Finally, we point out that the performance of invex regularizers in Eqs. (7), (8), (9), and (10) can be justified under the framework of re-weighted $\ell_1$-norm minimization (see [5]), which enhances the performance of just $\ell_1$-norm minimization.

## 2 Proof of Lemma 2

*Proof.* To prove this theorem, we show that for each $\boldsymbol{x} \in \mathbb{R}^n$ such that $\boldsymbol{0} \in \partial h(\boldsymbol{x})$ where $h(\boldsymbol{x}) = f(\boldsymbol{H}\boldsymbol{x} - \boldsymbol{v})$ is a global minimizer. Observe that

$$\partial h(\boldsymbol{x}) = \{\nabla h(\boldsymbol{x})\} = \left\{ \boldsymbol{H}^T \nabla f(\boldsymbol{H}\boldsymbol{x} - \boldsymbol{v}) \right\}. \tag{9}$$

Take $\boldsymbol{x}^* \in \mathbb{R}^n$ such that $\boldsymbol{0} \in \partial h(\boldsymbol{x}^*)$, then since $\boldsymbol{H}$ is a full row-rank matrix (equivalently $\boldsymbol{H}^T$ full col-rank matrix) from Eq. (9) we have

$$\nabla h(\boldsymbol{x}^*) = \boldsymbol{H}^T \nabla f(\boldsymbol{H}\boldsymbol{x}^* - \boldsymbol{v}) = \boldsymbol{0} \leftrightarrow \nabla f(\boldsymbol{H}\boldsymbol{x}^* - \boldsymbol{v}) = \boldsymbol{0}. \tag{10}$$

The above equation means that each stationary point of $h(\boldsymbol{x})$ is found through the stationary points of $f(\boldsymbol{x})$. Thus, since $f$ is invex then $\boldsymbol{H}\boldsymbol{x}^* - \boldsymbol{v}$ is a global minimizer of $f$ i.e. $h$ is invex. $\qquad\square$

## 3 Proof of Theorem 3

In this appendix we seek to guarantee that the proximal operator of the functions in Table 2 are invex. We point out that, since the proximal of the regularizers in Table 2 is the sum of a scalar function applied to each entry of a vector, then it is enough to analyze the scalar function to determine the invexity of the proximal.

### 3.1 Invexity proofs of the proximal operators

In the following we provide the proof for the first statement in Theorem 3.

**Eq.** (6)

*Proof.* Let $h(w)$ be a function defined, for $p \in (0, 1)$, as

$$h(w) = (|w| + \epsilon)^p + \frac{1}{2}(w - u)^2, \tag{11}$$

for fixed $u \in \mathbb{R}$, and $\epsilon \geq (p(1-p))^{\frac{1}{2-p}}$. Then, we seek to show that the second derivate of $h(w)$ with respect to $w$ for $w \neq 0$ is non-negative. Observe that,

$$h''(w) = \frac{p(p-1)}{(|w|+\epsilon)^{2-p}} + 1. \tag{12}$$

From Eq. (12) we have that $(|w|+\epsilon)^{2-p}$ is a positive increasing function since $2-p > 1$. This implies that to show $h''(w)$ is non-negative for all $w \in \mathbb{R}$ we need to analyze only when $w = 0$. Therefore, $\frac{p(p-1)}{(|w|+\epsilon)^{2-p}} \in [-1, 0)$ for all $w \in \mathbb{R}$, because $p(p-1) < 0$ and $\epsilon \geq (p(1-p))^{\frac{1}{2-p}}$. Thus, $h''(w)$ is non-negative, leading to the invexity of $h(w)$ (i.e. $h''(w)$ positive implies convexity). $\square$

**Eq. (7)**

*Proof.* Take $h(w) = \log(1 + |w|) + \frac{1}{2}(w-u)^2$ for fixed $u \in \mathbb{R}$. Observe that the second derivative of $h(w)$, for $w \neq 0$ is given by

$$h''(w) = \frac{-1}{(1+|w|)^2} + 1. \tag{13}$$

Then, since $(1 + |w|)^2 \geq 1$ for all $w$, this implies that $\frac{-1}{(1+|w|)^2} \in [-1, 0)$. Thus, $h''(w)$ is non-negative, leading to the invexity of $h(w)$. $\square$

**Eq. (8)**

*Proof.* Take $h(w) = \frac{|w|}{2+2|w|} + \frac{1}{2}(w-u)^2$ for fixed $u \in \mathbb{R}$. We will use the same argument as in previous cases. Then, for $w \neq 0$ notice that the second derivative of $h(w)$ is given by

$$h''(w) = \frac{-1}{(1+|w|)^3} + 1. \tag{14}$$

Then, from the above equation it is clear that $\frac{-1}{(1+|w|)^3} \in [-1, 0)$ for all $w \in \mathbb{R}$. Thus, $h''(w)$ is non-negative, leading to the invexity of $h(w)$. $\square$

**Eq. (9)**

*Proof.* Take $h(w) = \frac{w^2}{1+w^2} + \frac{1}{2}(w-u)^2$, for fixed $u \in \mathbb{R}$. Then, notice that the second derivative of $h(w)$ is given by

$$h''(w) = \frac{2 - 6w^2}{(1+w^2)^3} + 1. \tag{15}$$

Then, we show that $s(w) = \frac{2-6w^2}{(1+w^2)^3} \geq -1$, by determining its extreme values. Observe that

$$s'(w) = \frac{24w(w^2-1)}{(1+w^2)^3} = 0, \tag{16}$$

only when $w = 0, 1, -1$. It is clear that the maximum value of $s(w)$ is attained when $w = 0$, i.e. $s(w) = 2$. And, its minimum value is achieved when $w = -1$, that is $s(1) = s(-1) = \frac{-1}{2}$. Thus, since $s(w) \geq -1$ then $h(w)$ is invex. $\square$

**Eq. (10)**

*Proof.* Take $h(w) = \log(1 + |w|) - \frac{|w|}{2+2|w|} + \frac{1}{2}(w-u)^2$, for fixed $u \in \mathbb{R}$. Then, for $w \neq 0$ notice that the second derivative of $h(w)$ is given by

$$h''(w) = \frac{-|w|}{(1+|w|)^3} + 1. \tag{17}$$

Then, from the above equation it is clear that $\frac{-|w|}{(1+|w|)^3} \in [-1, 1]$, which implies that $h''(w)$ is non-negative for any $w$. Thus, $h(w)$ is invex. $\square$

## 3.2 The resolvent of proximal operator only has global optimizers

*Proof.* Now we proof the second part of Theorem 3. From the previous analysis on each proximal operator, we have that $h(\boldsymbol{x})$ is an convex (therefore invex) function, then Theorem 2 states that any global minimizer $\boldsymbol{y}$ of $h$ satisfies that $\boldsymbol{0} \in \partial h(\boldsymbol{y})$. This condition implies that $\boldsymbol{0} \in \partial g(\boldsymbol{y}) + (\boldsymbol{y} - \boldsymbol{v})$, from which we obtain that $\boldsymbol{y} \in (\partial g + \mathbf{I})^{-1}(\boldsymbol{v})$. Thus, we have that $\mathbf{prox}_g(\boldsymbol{v}) = (\partial g + \mathbf{I})^{-1}(\boldsymbol{v})$ from which the result holds. $\qquad\square$

## 3.3 Numerical Analysis of Proximal

In this section we present additional numerical analysis on the proximal of invex regularizers listed in Table 2. We start by providing a visual comparison between the one-dimensional version of $\ell_1$-norm and the invex regularizers in Eqs. (6), (7), (8), (9), and (10). This comparison is reported in Fig. 1. From this illustration it is easy to conclude why Eqs. (6), (7), (8), (9), and (10) are non-convex.

To complement the comparison between convex and invex regularizers, we present a graphical validation of the theoretical result in Theorem 3. To that end, we illustrate also in Fig. 1 the landscape of function $h(\boldsymbol{x}) = g(\boldsymbol{x}) + \frac{1}{2}\|\boldsymbol{x} - \boldsymbol{u}\|_2^2$ where $\boldsymbol{x}$ is a vector of two dimensions $\boldsymbol{x} = [x_1, x_2]^T$, $\boldsymbol{u} = [1, 1]^T$, with $g(\boldsymbol{x})$ taking the form of all invex regularizers in Eqs. (6), (7), (8), (9), and (10). From these results, it is clear that the level curves are concentric convex sets which confirms that $h(\boldsymbol{x})$ is convex (therefore invex), as stated in Theorem 3.

Lastly, the running time to compute the proximal of invex regularizers is also an important aspect to compare with its convex competitor i.e. $\ell_1$-norm. The reason for this, is because it is desire to improve imaging quality keeping the same computational complexity to obtain it. Therefore, the following Table 1 reports the running time to compute the proximal (in GPU) of Eqs. (6), (7), (8),

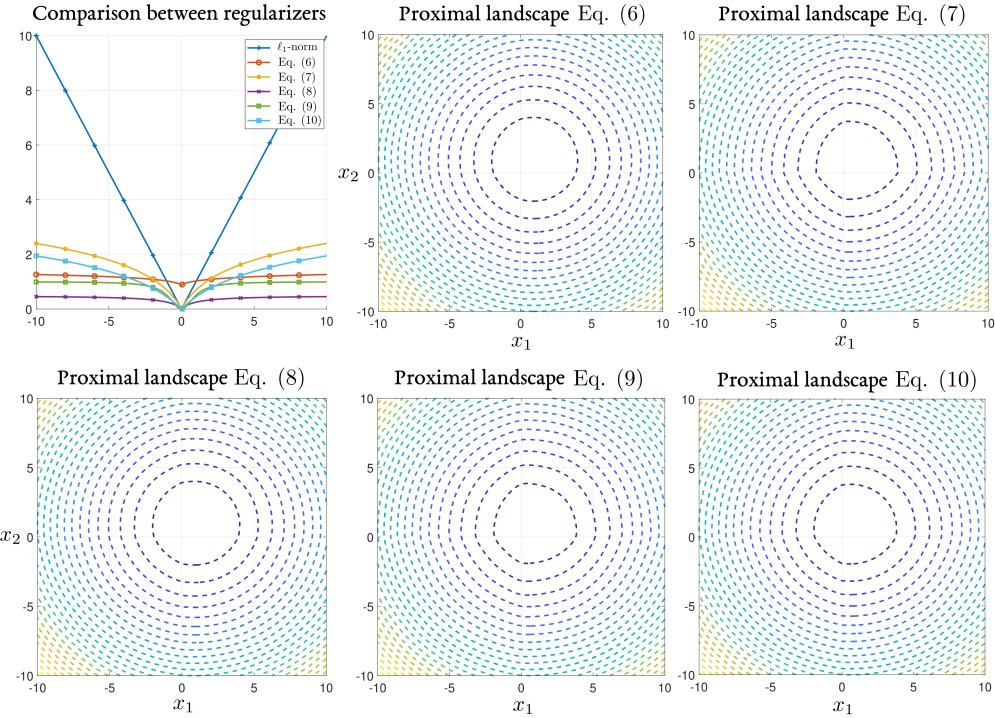

Figure 1: Here we present a visual comparison between the one-dimensional version of $\ell_1$-norm and the invex regularizers in Eqs. (6), (7), (8), (9), and (10). For Eq. (6) we select $p = 0.5$, and $\epsilon = (p(1-p))^{\frac{1}{2-p}}$. We also report the landscape of function $h(\boldsymbol{x}) = g(\boldsymbol{x}) + \frac{1}{2}\|\boldsymbol{x} - \boldsymbol{u}\|_2^2$ where $\boldsymbol{x}$ is a vector of two dimensions $\boldsymbol{x} = [x_1, x_2]^T$, $\boldsymbol{u} = [1, 1]^T$, and for all invex regularizers in Eqs. (6), (7), (8), (9), and (10). It is clear that the level curves are concentric convex sets which confirms that $h(\boldsymbol{x})$ is convex (therefore invex), as stated in Theorem 3.

(9), and (10) for an image of $2048 \times 2048$ pixels. Observe that Table 1 suggests that computing the proximal of the $\ell_1$-norm is faster than the proximal of invex regularizers. However, this difference is given in milliseconds making it negligible in practice.

Table 1: Time to compute the proximal for all invex and convex regularizers, of an image with $2048 \times 2048$ pixels. The reported time is the averaged over 256 trials. For Eq. (6) we select $p = 0.5$, and $\epsilon = (p(1-p))^{\frac{1}{2-p}}$.

|      | Eq. (6) | Eq. (7) | Eq. (8) | Eq. (9) | Eq. (10) | $\ell_1$-norm |
|------|---------|---------|---------|---------|----------|---------------|
| Time | $1.47ms$ | $0.63ms$ | $2.8ms$ | $4.7ms$ | $2.4ms$ | $0.66ms$ |

## 4    Solutions to the Proximal Operator in Eq. (11)

In this section we present the proximal operator for the functions in Eqs. (6)-(10) summarized in Table 2. In the case of Eq. (6) its proximal operator was calculated in [6]. We recall that the analysis for Eq. (6) is valid with and without the constant $\epsilon$. We prefer to add $\epsilon$ in order to formally satisfy the Lipschitz continuity as in Definition 1. Moreover, for functions in Eqs. (7)-(10) we present how to estimate their proximal operator them in the following.

**Proximal of Eq. (7)**    Consider $h(w) = \lambda \log(1 + |w|) + \frac{1}{2}(w - u)^2$ for $\lambda \in (0, 1]$, and fixed $u \in \mathbb{R}$. We note first that we only consider $w's$ for which $\text{sign}(w) = \text{sign}(u)$, otherwise $h(w) = \lambda \log(1 + |w|) + \frac{1}{2}w^2 + |u||w| + \frac{1}{2}u^2$ which is clearly minimized at $w = 0$. Then, since with $\text{sign}(w) = \text{sign}(u)$ we have $(w - u)^2 = (|w| - |u|)^2$, we replace $u$ with $|u|$ and take $w \geq 0$. As $h(w)$ is differentiable for $w > 0$, re-arranging $h'(w) = 0$ gives

$$\psi_\lambda(w) \triangleq \frac{\lambda}{1 + w} + w = |u|. \tag{18}$$

Observe that $\psi'_\lambda(w)$ is always positive then it means that $\psi_\lambda(w)$ is monotonically increasing. Thus, the equation $\psi_\lambda(w) = |u|$ has unique solution i.e. at some point the quality holds. Thus, solving $\psi_\lambda(w) = |u|$ is equivalent to

$$w^2 + (1 - |u|)w + \lambda - |u| = 0. \tag{19}$$

It is easy to verify that the solution to Eq. (19) that returns the minimum value of $h(w)$ is given by $w = \frac{|u| - 1 + \sqrt{(|u| + 1)^2 - 4\lambda}}{2}$ when $(|u| + 1)^2 \geq 4\lambda$, and 0 otherwise.

**Proximal of Eq. (8)**    Consider $h(w) = \lambda \frac{|w|}{2 + 2|w|} + \frac{1}{2}(w - u)^2$ for $\lambda \in (0, 1]$, and fixed $u \in \mathbb{R}$. We note first that we only consider $w's$ for which $\text{sign}(w) = \text{sign}(u)$, otherwise $h(w) = \lambda \frac{|w|}{2 + 2|w|} + \frac{1}{2}w^2 + |u||w| + \frac{1}{2}u^2$ which is clearly minimized at $w = 0$. Then, since with $\text{sign}(w) = \text{sign}(u)$ we have $(w - u)^2 = (|w| - |u|)^2$, we replace $u$ with $|u|$ and take $w \geq 0$. As $h(w)$ is differentiable for $w > 0$, re-arranging $h'(w) = 0$ gives

$$\psi_\lambda(w) \triangleq \frac{\lambda}{2(1 + w)^2} + w = |u|. \tag{20}$$

Observe that $\psi'_\lambda(w)$ is always positive then it means that $\psi_\lambda(w)$ is monotonically increasing. Thus, the equation $\psi_\lambda(w) = |u|$ has unique solution i.e. at some point the quality holds. Thus, solving $\psi_\lambda(w) = |u|$ is equivalent to

$$2w^3 + (4 - 2|u|)w^2 + (2 - 4|u|)w + \lambda - 2|u| = 0. \tag{21}$$

Equation (21) is easily solved using traditional python packages[2].

---

[2]Example of Python function to solve Eq. (21) at `https://numpy.org/doc/stable/reference/generated/numpy.roots.html`.

Table 2: Invex regularization functions from Table 2 and their corresponding proximity operator ($\lambda \in (0, 1]$ is a thresholding parameter).

| Ref | Invex function | Proximal operator |
|-----|----------------|-------------------|
| [6] | $g_\lambda(x) = \lambda|x|^p, p \in (0,1), x \neq 0.$ | $\text{Prox}_{g_\lambda}(t) = \begin{cases} 0 & |t| < \tau \\ \{0, \text{sign}(t)\beta\} & |t| = \tau \\ \text{sign}(t)y & |t| > \tau \end{cases}$ where $\beta = [2\lambda(1-p)]^{1/(2-p)}, \tau = \beta + \lambda p \beta^{p-1}$, $h(y) = \lambda p y^{p-1} + y - |t| = 0, y \in [\beta, |t|]$ |
| - | $g_\lambda(x) = \lambda \log(1 + |x|)$ | $\text{Prox}_{g_\lambda}(t) = \begin{cases} 0 & (|t|+1)^2 < 4\lambda \\ \text{sign}(t)\beta & \beta \geq 0 \\ 0 & \text{otherwise} \end{cases}$ where $\beta = \frac{|t|-1+\sqrt{(|t|+1)^2-4\lambda}}{2}$. |
| - | $g_\lambda(x) = \lambda \frac{|x|}{2+2|x|}$ | $\text{Prox}_{g_\lambda}(t) = \begin{cases} 0 & |t| = 0 \\ \text{sign}(t)\beta & \text{otherwise} \end{cases}$ where $2\beta^3 + (4-2|t|)\beta^2 + (2-4|t|)\beta + \lambda - 2|t| = 0, \beta > 0$, and closest to $|t|$. |
| - | $g_\lambda(x) = \lambda \frac{x^2}{1+x^2}$ | $\text{Prox}_{g_\lambda}(t) = \begin{cases} 0 & |t| = 0 \\ \text{sign}(t)\beta & \text{otherwise} \end{cases}$ where $\beta^5 - |t|\beta^4 + 2\beta^3 - 2|t|\beta^2 + (1+2\lambda)\beta - |t| = 0, \beta > 0$, and closest to $|t|$ |
| - | $g_\lambda(x) = \lambda\left(\log(1+|x|) - \frac{|x|}{2+2|x|}\right)$ | $\text{Prox}_{g_\lambda}(t) = \begin{cases} 0 & |t| = 0 \\ \text{sign}(t)\beta & \text{otherwise} \end{cases}$ where $2\beta^3 + (4-2|t|)\beta^2 + (2\lambda + 2 - 4|t|)\beta + \lambda - 2|t| = 0, \beta > 0$, and closest to $|t|$. |

**Proximal of Eq. (9)** Consider $h(w) = \lambda\frac{w^2}{1+w^2} + \frac{1}{2}(w-u)^2$ for $\lambda \in (0, 1]$, and fixed $u \in \mathbb{R}$. We note first that we only consider $w's$ for which $\text{sign}(w) = \text{sign}(u)$, otherwise $h(w) = \lambda\frac{w^2}{1+w^2} + \frac{1}{2}w^2 + |u||w| + \frac{1}{2}u^2$ which is clearly minimized at $w = 0$. Then, since with $\text{sign}(w) = \text{sign}(u)$ we have $(w-u)^2 = (|w|-|u|)^2$, we replace $u$ with $|u|$ and take $w \geq 0$. As $h(w)$ is differentiable for $w > 0$, re-arranging $h'(w) = 0$ gives

$$\psi_\lambda(w) \triangleq \frac{2\lambda w}{(1+w^2)^2} + w = |u|. \tag{22}$$

Observe that $\psi'_\lambda(w)$ is always positive then it means that $\psi_\lambda(w)$ is monotonically increasing. Thus, the equation $\psi_\lambda(w) = |u|$ has unique solution i.e. at some point the quality holds. Thus, solving $\psi_\lambda(w) = |u|$ is equivalent to

$$w^5 - |u|w^4 + 2w^3 - 2|u|w^2 + (1+2\lambda)w - |u| = 0. \tag{23}$$

Equation (23) is easily solved using traditional python packages.

**Proximal of Eq. (10)** Consider $h(w) = \lambda\left(\log(1+|w|) - \frac{|w|}{2+2|w|}\right) + \frac{1}{2}(w-u)^2$ for $\lambda \in (0, 1]$, and fixed $u \in \mathbb{R}$. We note first that we only consider $w's$ for which $\text{sign}(w) = \text{sign}(u)$, otherwise $h(w) = \lambda\left(\log(1+|w|) - \frac{|w|}{2+2|w|}\right) + \frac{1}{2}w^2 + |u||w| + \frac{1}{2}u^2$ which is clearly minimized at $w = 0$. Then, since with $\text{sign}(w) = \text{sign}(u)$ we have $(w-u)^2 = (|w|-|u|)^2$, we replace $u$ with $|u|$ and take $w \geq 0$. As $h(w)$ is differentiable for $w > 0$, re-arranging $h'(w) = 0$ gives

$$\psi_\lambda(w) \triangleq \lambda\frac{2w+1}{2(1+w)^2} + w = |u|. \tag{24}$$

Observe that $\psi'_\lambda(w)$ is always positive then it means that $\psi_\lambda(w)$ is monotonically increasing. Thus, the equation $\psi_\lambda(w) = |u|$ has unique solution i.e. at some point the quality holds. Thus, solving

$\psi_\lambda(w) = |u|$ is equivalent to

$$2w^3 + (4 - 2|u|)w^2 + (2\lambda + 2 - 4|u|)w + \lambda - 2|u| = 0. \tag{25}$$

Equation (25) is easily solved using traditional python packages.

# 5 Proof of Theorem 4

We split the proof of Theorem 4 into two parts. First part we focus our analysis on functions in Eqs. (7),(8),(10) and second part Eq. (9) (extra mild conditions are needed). Recall we skipped Eq. (6).

## 5.1 Part one

We particularized [7, Theorem 1] in order to prove that whenever the $\ell_1$-norm solution of optimization problem in Eq. (12) is unique, then Eq. (12) when $g(x)$ satisfies the following definition has the same global optima.

**Definition 1.** (*Sparseness measure* [7]) Let $g : \mathbb{R}^n \to \mathbb{R}$ such that $g(w) = \sum_{i=1}^{n} r(w[i])$, where $r : [0, \infty) \to [0, \infty)$ and increasing. If $r$, not identically zero, with $r(0) = 0$ such that $r(t)/t$ is non-increasing on $(0, \infty)$, then $g(x)$ is said to be a *sparseness measure*.

Now we present the particular version in [7, Theorem 1] as follows.

**Lemma 1.** Assume $Hx = b$, where $x \in \mathbb{R}^n$ is $k$-sparse, the matrix $H \in \mathbb{R}^{m \times n}$ ($m < n$) with $\ell_2$-normalized columns that satisfies RIP for any $2k$-sparse vector, with $\delta_{2k} < \frac{1}{3}$, and $b \in \mathbb{R}^m$ is a noiseless measurements data vector. If $g(x)$ in Eqs. (7),(8), (10) satisfies Definition 1, then $x$ is exactly recovered by solving Eq. (12) i.e. only global optimizers exists.

In the following we prove functions in Eqs. (7),(8), and (10) satisfy Definition 1, and we proceed by cases.

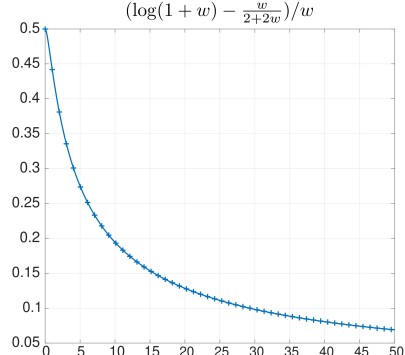

*Proof.* **Eq. (7):** Take $g(w) = \log(1 + |w|)$ for any $w \in \mathbb{R}$. It is trivial to see that $g(0) = 0$, and that $g(w)$ it is not identically zero. Then, we just need to show that $g(w)/w$ is non-increasing on $(0, \infty)$. Define $r(w) = \frac{\log(1+w)}{w}$. Observe that the derivative of $r(w)$ is given by $r'(w) = \frac{\frac{w}{1+w} - \log(1+w)}{w^2}$, for $w \in (0, \infty)$. Since $\frac{w}{1+w} - \log(1 + w) < 0$, then $r'(w) < 0$ leads to conclude that $g(w)/w$ is non-increasing on $(0, \infty)$.

Figure 2: Plot of $g(w)/w$ for $g(w)$ being Eq. (10) and $w > 0$ to check that $g(w)/w$ is non-increasing on $(0, \infty)$.

**Eq. (8):** Take $g(w) = \frac{|w|}{2 + 2|w|}$ for any $w \in \mathbb{R}$. It is trivial to see that $g(0) = 0$, and that $g(w)$ it is not identically zero. Then, we just need to show that $g(w)/w$ is non-increasing on $(0, \infty)$. Define $r(w) = \frac{w}{2w + 2w^2} = \frac{1}{2+2w}$. Then, it is clear to conclude that $g(w)/w$ is non-increasing on $(0, \infty)$.

**Eq. (10):** Take $g(w) = \log(1 + |w|) - \frac{|w|}{2 + 2|w|}$ for any $w \in \mathbb{R}$. It is trivial to see that $g(0) = 0$, and that $g(w)$ it is not identically zero. Then, we just need to show that $g(w)/w$ is non-increasing on $(0, \infty)$. For easy of exposition we present in Figure 2 the plot of $g(w)/w$. Then it is clear that $g(w)/w$ is non-increasing on $(0, \infty)$. $\square$

## 5.2 Part two

For this second part we appeal to a generalized result of [7, Theorem 1] presented in [8, Theorem 3.10]. To exploit this generalized theorem we introduce the following definition.

**Definition 2.** (*Admissible sparseness measure* [8]) A function $g : \mathbb{R}^n \to \mathbb{R}$ such that $g(w) = \sum_{i=1}^{n} r(w[i])$ is said to be an admissible sparseness measure if

- $r(0) = 0$, and g even on $\mathbb{R}$,

- $r$ is continuous on $\mathbb{R}$, and strictly increasing and strictly concave on $\mathbb{R}$.

Based on the above definition we particularized [8, Theorem 3.10] in the lemma below in order to prove the solution of optimization problem in Eq. (12) is unique, when functions in Eq. (9) are used under some mild conditions.

**Lemma 2.** ([8, Theorem 3.10]) Assume $\boldsymbol{Hx} = \boldsymbol{b}$, where $\boldsymbol{x} \in \mathbb{R}^n$ is $k$-sparse, the matrix $\boldsymbol{H} \in \mathbb{R}^{m \times n}$ $(m < n)$ with $\ell_2$-normalized columns that satisfies RIP for $\delta_s \in (0, 1)$ with $s \geq 2k$, and $\boldsymbol{b} \in \mathbb{R}^m$ is a noiseless measurements data vector. Define $\beta_1, \beta_2 > 0$ to be the lower and upper bound of magnitudes of non-zero entries of feasible vectors of Eq. (12) (their existence if guaranteed [8]). If $kr(2\beta_2) < (s + k - 1)r(\beta_1)$, then $\boldsymbol{x}$ is exactly recovered by solving Eq. (12) i.e. only global optimizers exists.

In the following we prove functions from Eq. (9) in Table 2 are able to exactly recover the signal $\boldsymbol{x}$ under some mild conditions.

*Proof.* **Eq. (9):** Take $r(w) = \frac{w^2}{1+w^2}$ for any $w \in \mathbb{R}$. It is trivial to see that $r(0) = 0$, to check that it is even, continuous, and strictly increasing. Observe that the second derivative of $r(w)$ is given by $r''(w) = \frac{2-6w^2}{(1+w^2)^2}$. Then it is clear to conclude that $r(w)$ is strictly concave when $w > \frac{1}{3}$. Then, in order to have the chance to exactly recover the signal $\boldsymbol{x}$ we need to assume that the lower bound of magnitudes of non-zero entries of feasible vectors is $\beta_1 > \frac{1}{3}$. Without loss of generality we assume $\boldsymbol{x}$ is a normalized signal (in practical imaging applications $\boldsymbol{x}$ is always normalized). Then, we take $\beta_1 = 0.5$, and $\beta_2 = 1.0$. In addition, assuming $\boldsymbol{H}$ satisfies RIP when $s \geq 4k + 2$, with $\delta_s \in (0, 1)$, it is numerically easy to verified that $kr(2\beta_2) < (s + k - 1)r(\beta_1)$. $\qquad \square$

## 6 Proof of Lemma 3

Before proving Lemma 3 we consider two definitions in the following which the loss function $F(\boldsymbol{x}) = f(\boldsymbol{x}) + \lambda g(\boldsymbol{x})$ in Eq. (12) satisfies. Recall that $\lambda \in (0, 1]$.

**Definition 3.** A function $h : \mathbb{R}^n \to (-\infty, \infty]$ is said to be proper if dom $h \neq \emptyset$, where dom $= \{\boldsymbol{x} \in \mathbb{R}^n : h(\boldsymbol{x}) < \infty\}$.

Since we are assuming the sensing matrix $\boldsymbol{H}$ satisfies RIP it guarantees the existence of a solution to Eq. (12) implying that dom $F \neq \emptyset$. Thus, $F(\boldsymbol{x})$ in Eq. (12) satisfies the above definition because.

**Definition 4.** A function $h : \mathbb{R}^n \to \mathbb{R}$ is coercive, if $h$ is bounded from below and $h(\boldsymbol{x}) \to \infty$ when $\|\boldsymbol{x}\|_2 \to \infty$.

Considering that the list of invex functions in Table 2, and $f(\boldsymbol{x}) = \|\boldsymbol{Hx} - \boldsymbol{v}\|_2^2$ (for fix $\boldsymbol{v}$ and $\boldsymbol{H}$ satisfying RIP) are positive, then the loss function $F(\boldsymbol{x})$ satisfies $F(\boldsymbol{x}) \geq 0$. The second part of the coercive definition is trivially guaranteed since $\boldsymbol{H}$ satisfies RIP, otherwise we will be denying the existence of a global solution to Eq. (12) which is a contradiction.

Now we proceed to prove Lemma 3.

*Proof.* Line 6 in Algorithm 1 is given by

$$\boldsymbol{v}^{(t+1)} = \underset{\boldsymbol{x} \in \mathbb{R}^n}{\arg\min} \ \left\langle \nabla f(\boldsymbol{x}^{(t)}), \boldsymbol{x} - \boldsymbol{x}^{(t)} \right\rangle + \frac{1}{2\lambda\alpha_1}\|\boldsymbol{x} - \boldsymbol{x}^{(t)}\|_2^2 + g(\boldsymbol{x}). \qquad (26)$$

We write equal in the above equation because the proximal in Line 6 is invex therefore it always map to a global optimizer. So from Eq. (26) we have

$$\left\langle \nabla f(\boldsymbol{x}^{(t)}), \boldsymbol{v}^{(t+1)} - \boldsymbol{x}^{(t)} \right\rangle + \frac{1}{2\lambda\alpha_1}\|\boldsymbol{v}^{(t+1)} - \boldsymbol{x}^{(t)}\|_2^2 + g(\boldsymbol{v}^{(t+1)}) \leq g(\boldsymbol{x}^{(t)}). \qquad (27)$$

From the Lipschitz continuous of $\nabla f$ and Eq. (27) we have

$$F(\boldsymbol{v}^{(t+1)}) \leq g(\boldsymbol{v}^{(t+1)}) + f(\boldsymbol{x}^{(t)}) + \left\langle \nabla f(\boldsymbol{x}^{(t)}), \boldsymbol{v}^{(t+1)} - \boldsymbol{x}^{(t)} \right\rangle + \frac{L}{2} \|\boldsymbol{v}^{(t+1)} - \boldsymbol{x}^{(t)}\|_2^2$$

$$\leq g(\boldsymbol{x}^{(t)}) - \left\langle \nabla f(\boldsymbol{x}^{(t)}), \boldsymbol{v}^{(t+1)} - \boldsymbol{x}^{(t)} \right\rangle - \frac{1}{2\lambda\alpha_1} \|\boldsymbol{v}^{(t+1)} - \boldsymbol{x}^{(t)}\|_2^2$$

$$+ f(\boldsymbol{x}^{(t)}) + \left\langle \nabla f(\boldsymbol{x}^{(t)}), \boldsymbol{v}^{(t+1)} - \boldsymbol{x}^{(t)} \right\rangle + \frac{L}{2} \|\boldsymbol{v}^{(t+1)} - \boldsymbol{x}^{(t)}\|_2^2$$

$$= F(\boldsymbol{x}^{(t)}) - \left( \frac{1}{2\lambda\alpha_1} - \frac{L}{2} \right) \|\boldsymbol{v}^{(t+1)} - \boldsymbol{x}^{(t)}\|_2^2. \tag{28}$$

If $F(\boldsymbol{z}^{(t+1)}) \leq F(\boldsymbol{v}^{(t+1)})$, then

$$\boldsymbol{x}^{(t+1)} = \boldsymbol{z}^{(t+1)}, F(\boldsymbol{x}^{(t+1)}) = F(\boldsymbol{z}^{(t+1)}) \leq F(\boldsymbol{v}^{(t+1)}). \tag{29}$$

If $F(\boldsymbol{z}^{(t+1)}) > F(\boldsymbol{v}^{(t+1)})$, then

$$\boldsymbol{x}^{(t+1)} = \boldsymbol{v}^{(t+1)}, F(\boldsymbol{x}^{(t+1)}) = F(\boldsymbol{v}^{(t+1)}). \tag{30}$$

From Eqs. (28), (29) and (30) we have

$$F(\boldsymbol{x}^{(t+1)}) \leq F(\boldsymbol{v}^{(t+1)}) \leq F(\boldsymbol{x}^{(t)}). \tag{31}$$

So

$$F(\boldsymbol{x}^{(t+1)}) \leq F(\boldsymbol{x}^{(1)}), F(\boldsymbol{v}^{(t+1)}) \leq F(\boldsymbol{x}^{(1)}), \tag{32}$$

for all $t$. Recall that we consider the estimation of $\boldsymbol{z}^{(t+1)}$ unique because it is performed through the proximal of $g(\boldsymbol{x})$ which always map to a global optimizer.

Observe that from Eq. (31) was concluded that $F(\boldsymbol{x}^{(t)})$ is nonincreasing then for all $t > 1$ we have $F(\boldsymbol{x}^{(t)}) \leq F(\boldsymbol{x}^{(1)})$ and therefore $\boldsymbol{x}^{(t)} \in \{\boldsymbol{w} : F(\boldsymbol{w}) \leq F(\boldsymbol{x}^{(1)})\}$ (known as level sets). Since $F(\boldsymbol{x})$ is coercive then all its level sets are bounded. Then we know that $\{\boldsymbol{x}^{(t)}\}$, and $\{\boldsymbol{v}^{(t)}\}$ are also bounded. Thus $\{\boldsymbol{x}^{(t)}\}$ has accumulation points. Let $\boldsymbol{x}^*$ be any accumulation point of $\{\boldsymbol{x}^{(t)}\}$, say a subsequence satisfying $\{\boldsymbol{x}^{(t_j+1)}\} \to \boldsymbol{x}^*$ as $j \to \infty$. Let $F^*$ be $\lim_{j \to \infty} F(\boldsymbol{x}^{(t_j+1)}) = F(\boldsymbol{x}^*) = F^*$. The existence of this limit is guaranteed since $f$ is continuously differentiable. Then, from Eq. (28) we have

$$\left( \frac{1}{2\lambda\alpha_1} - \frac{L}{2} \right) \|\boldsymbol{v}^{(t+1)} - \boldsymbol{x}^{(t)}\|_2^2 \leq F(\boldsymbol{x}^{(t)}) - F(\boldsymbol{v}^{(t+1)}) \leq F(\boldsymbol{x}^{(t)}) - F(\boldsymbol{x}^{(t+1)}). \tag{33}$$

Summing over $t = 1, 2, \ldots, \infty$, we have

$$\left( \frac{1}{2\lambda\alpha_1} - \frac{L}{2} \right) \sum_{t=1}^{\infty} \|\boldsymbol{v}^{(t+1)} - \boldsymbol{x}^{(t)}\|_2^2 \leq F(\boldsymbol{x}^{(1)}) - F^* < \infty. \tag{34}$$

From $\alpha_1 < \frac{1}{L}$ we have

$$\|\boldsymbol{v}^{(t+1)} - \boldsymbol{x}^{(t)}\|_2^2 \to 0, \text{ as } t \to \infty. \tag{35}$$

From the optimality condition of Eq. (26) we have

$$\boldsymbol{0} \in \nabla f(\boldsymbol{x}^{(t)}) + \frac{1}{\lambda\alpha_1} (\boldsymbol{v}^{(t+1)} - \boldsymbol{x}^{(t)}) + \partial g(\boldsymbol{v}^{(t+1)})$$

$$= \nabla f(\boldsymbol{x}^{(t)}) + \nabla f(\boldsymbol{v}^{(t+1)}) - \nabla f(\boldsymbol{v}^{(t+1)}) + \frac{1}{\lambda\alpha_1} (\boldsymbol{v}^{(t+1)} - \boldsymbol{x}^{(t)}) + \partial g(\boldsymbol{v}^{(t+1)}). \tag{36}$$

So we have

$$-\nabla f(\boldsymbol{x}^{(t)}) + \nabla f(\boldsymbol{v}^{(t+1)}) - \frac{1}{\lambda\alpha_1} (\boldsymbol{v}^{(t+1)} - \boldsymbol{x}^{(t)}) \in \partial F(\boldsymbol{v}^{(t+1)}), \tag{37}$$

and

$$\left\| \nabla f(\boldsymbol{x}^{(t)}) - \nabla f(\boldsymbol{v}^{(t+1)}) + \frac{1}{\lambda\alpha_1} (\boldsymbol{v}^{(t+1)} - \boldsymbol{x}^{(t)}) \right\|_2 \leq \left( \frac{1}{\lambda\alpha_1} + L \right) \|\boldsymbol{v}^{(t+1)} - \boldsymbol{x}^{(t)}\|_2 \to 0, \tag{38}$$

as $t \to \infty$.

From Eq. (35) we have $\boldsymbol{v}^{(t_j+1)} \to \boldsymbol{x}^*$ as $j \to \infty$. From Eq. (26) we have

$$\left\langle \nabla f(\boldsymbol{x}^{(t_j)}), \boldsymbol{v}^{(t_j+1)} - \boldsymbol{x}^{(t_j+1)} \right\rangle + \frac{1}{2\lambda\alpha_1}\|\boldsymbol{v}^{(t_j+1)} - \boldsymbol{x}^{(t_j)}\|_2^2 + g(\boldsymbol{v}^{(t_j+1)})$$

$$\leq \left\langle \nabla f(\boldsymbol{x}^{(t_j)}), \boldsymbol{x}^* - \boldsymbol{x}^{(t_j)} \right\rangle + \frac{1}{2\lambda\alpha_1}\|\boldsymbol{x}^* - \boldsymbol{x}^{(t_j)}\|_2^2 + g(\boldsymbol{x}^*) \tag{39}$$

So

$$\limsup_{j \to \infty} \; g(\boldsymbol{v}^{(t_j+1)}) \leq g(\boldsymbol{x}^*). \tag{40}$$

From the continuity assumption on $g$ we have $\liminf\limits_{j \to \infty} \; g(\boldsymbol{v}^{(t_j+1)}) \geq g(\boldsymbol{x}^*)$, then we conclude

$$\lim_{j \to \infty} \; g(\boldsymbol{v}^{(t_j+1)}) = g(\boldsymbol{x}^*). \tag{41}$$

Because $f$ is continuously differentiable, we have $\lim\limits_{j \to \infty} \; F(\boldsymbol{v}^{(t_j+1)}) = F(\boldsymbol{x}^*)$. From $\{\boldsymbol{v}^{(t_j+1)}\} \to \boldsymbol{x}^*$, and Eq. (37) we have $\boldsymbol{0} \in \partial F(\boldsymbol{x}^*)$. Therefore, since $F(\boldsymbol{x})$ is invex according to Theorem 4 we have that the sequence $\{\boldsymbol{x}^{(t)}\}$ converges to a global minimizer of $F(\boldsymbol{x})$. $\qquad\square$

## 6.1 Numerical Validation of Lemma 3

To numerically validate the proof of Lemma 3 provided in the above section, we present Fig. 3. In this figure we are reporting the numerical convergence of Algorithm 1 for all invex regularizers to recover an image of size $256 \times 256$ from blurred data, as explained in Experiment 1 for the noiseless case. Specifically, Fig. 3(left) reports how the loss function $F(\boldsymbol{x}) = \ell_2 + \lambda g(\boldsymbol{x})$, analyzed in the above proof, is minimized along $T = 800$ iterations. This plot numerically validates the proof of Lemma 3. As a complement to this plot, Fig. 3(right) presents the running time of Algorithm 1 to perform $T = 800$ iterations for all invex regularizers and the $\ell_1$-norm. This second plot suggests that Algorithm 1 using the $\ell_1$-norm as regularizer requires 1.8 seconds less than its invex competitors to perform $T = 800$ iterations. We remark that this negligible difference is expected, since in Table 1 was concluded that the running time to compute the proximal operator for all invex differs in the order of milliseconds with the computation of the proximal of $\ell_1$-norm.

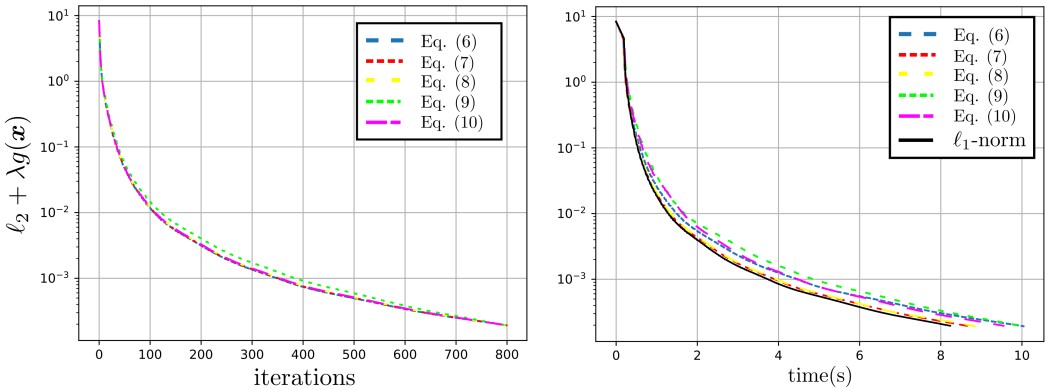

Figure 3: Numerical convergence of Algorithm 1 for all invex regularizers to recover an image of size $256 \times 256$ from blurred data, as explained in Experiment 1. (left) Minimization process of $F(\boldsymbol{x}) = \ell_2 + \lambda g(\boldsymbol{x})$ along $T = 800$ iterations. (right) running time of Algorithm 1 to perform $T = 800$ iterations for all invex regularizers and the $\ell_1$-norm.

# 7 Proof of Lemma 4

We proceed to prove this lemma by extending the mathematical analysis in [9] to invex functions. To that end, we recall some definitions and a classical result from monotone operator theory needed for the proof of this lemma as follows.

**Definition 5.** (*Nonexpansiveness*) An operator $F : \mathbb{R}^n \to \mathbb{R}^n$ is said to be nonexpansive if it is Lipschitz continuous as in Definition 1 with $L = 1$.

Based on the nonexpansiveness concept we give the following definition.

**Definition 6.** For a constant $\beta \in (0, 1)$ we say a function $G$ is $\beta$-average, if there exists a nonexpansive operator $F$ such that $G = (1 - \beta)\boldsymbol{I} + \beta F$

Now based on the concept of average operators we recall the following classical results.

**Lemma 3.** ([10, Proposition 4.44]) Let $G_1$ be $\beta_1$-averaged and $G_2$ be $\beta_2$-averaged. Then, the composite operator $G \triangleq G_2 \circ G_1$ is

$$\beta \triangleq \frac{\beta_1 + \beta_2 - 2\beta_1\beta_2}{1 - \beta_1\beta_2}, \tag{42}$$

averaged operator.

**Lemma 4.** Let $F$ be a $\beta$-average operator with $\beta \in (0, 1)$. Then

$$\|F(\boldsymbol{x}) - F(\boldsymbol{y})\|_2^2 \leq \|\boldsymbol{x} - \boldsymbol{y}\|_2^2 - \left(\frac{1 - \beta}{\beta}\right)\|\boldsymbol{x} - F(\boldsymbol{x}) - \boldsymbol{y} + F(\boldsymbol{y})\|_2^2. \tag{43}$$

Now we proceed to prove Lemma 4

*Proof.* Following the assumptions made in Lemma 4, we start this proof by noticing that for a differentiable invex function $f$ a point $\boldsymbol{x}$ is a global minimizer of $f$ according to Theorem 2 if

$$\boldsymbol{0} = \nabla f(\boldsymbol{x}) \leftrightarrow \boldsymbol{x} = (\boldsymbol{I} - \alpha\nabla f)(\boldsymbol{x}), \tag{44}$$

for non-zero $\alpha$. In other words, $\boldsymbol{x}$ is a minimizer of $f$ if and only if it is a fixed point of the mapping $\boldsymbol{I} - \alpha\nabla f$. This property of invex functions is what allows to extend the mathematical guarantees in [9] given only for convex functions. Now considering that $f$ is assumed to have Lipschitz continuous gradient with parameter $L$, then the operator $\boldsymbol{I} - \alpha\nabla f$ is Lipschitz with parameter $L_G = \max\{1, |1 - \alpha L|\}$ and therefore is nonexpansive for $\alpha \in (0, 2/L]$. So it is averaged for $\alpha \in (0, 2/L)$ since

$$\boldsymbol{I} - \alpha\nabla f = (1 - \kappa)\boldsymbol{I} + \kappa\left(\boldsymbol{I} - 2/L\nabla f\right), \tag{45}$$

where $\kappa = \alpha L/2 < 1$.

Assume the denoiser $d$ is $\kappa$-averaged and the operator $G_\alpha = \boldsymbol{I} - \alpha\nabla f$. Observe that $G_\alpha$ is $(\gamma L/2)$-averaged for any $\alpha \in (0, 2/L)$. From Lemma 3 their composition $P = d \circ G_\alpha$ is

$$\beta \triangleq \frac{\kappa + \gamma L/2 - 2\kappa\gamma L/2}{1 - \kappa\gamma L/2}, \tag{46}$$

averaged. Consider a single iteration $\boldsymbol{v}^+ = P(\boldsymbol{x})$, then we have for any $\boldsymbol{x}^* = P(\boldsymbol{x}^*)$ (fixed point) we have that

$$\begin{aligned}
\|\boldsymbol{v}^+ - \boldsymbol{x}^*\|_2^2 &= \|P(\boldsymbol{x}) - P(\boldsymbol{x}^*)\|_2^2 \\
&\leq \|\boldsymbol{x} - \boldsymbol{x}^*\|_2^2 - \left(\frac{1 - \beta}{\beta}\right)\|\boldsymbol{x} - P(\boldsymbol{x}) - \boldsymbol{x}^* + P(\boldsymbol{x}^*)\|_2^2 \\
&= \|\boldsymbol{x} - \boldsymbol{x}^*\|_2^2 - \left(\frac{1 - \beta}{\beta}\right)\|\boldsymbol{x} - P(\boldsymbol{x})\|_2^2,
\end{aligned} \tag{47}$$

where we used Lemma 4. From Line 6 in Algorithm 3 the iteration $t + 1$ and rearranging the terms, we obtain

$$\|\boldsymbol{x}^{(t)} - P(\boldsymbol{x}^{(t)})\|_2^2 \leq \left(\frac{\beta}{1 - \beta}\right)\left[\|\boldsymbol{x}^{(t)} - \boldsymbol{x}^*\|_2^2 - \|\boldsymbol{v}^{(t+1)} - \boldsymbol{x}^*\|_2^2\right]. \tag{48}$$

If $F(z^{(t+1)}) \leq F(v^{(t+1)})$, then

$$x^{(t+1)} = z^{(t+1)}, F(x^{(t+1)}) = F(z^{(t+1)}) \leq F(v^{(t+1)}). \tag{49}$$

If $F(z^{(t+1)}) > F(v^{(t+1)})$, then

$$x^{(t+1)} = v^{(t+1)}, F(x^{(t+1)}) = F(v^{(t+1)}). \tag{50}$$

From Eqs. (49) and (50) we have

$$F(x^{(t+1)}) \leq F(v^{(t+1)}) \leq F(x^{(t)}). \tag{51}$$

Observe that from Eq. (51) was concluded that $F(x^{(t)})$ is nonincreasing then for all $t > 1$ we have $F(x^{(t)}) \leq F(x^{(1)})$ and therefore $x^{(t)} \in \{w : F(w) \leq F(x^{(1)})\}$ (known as level sets). Since $F(x)$ is coercive then all its level sets are bounded (concluded from Appendix 6). Then we know that $\{x^{(t)}\}$, and $\{v^{(t)}\}$ are also bounded. Thus $\{x^{(t)}\}$ has accumulation points which guarantees the existence of $x^*$ implying that for a subsequence satisfying $\{x^{(t_j+1)}\} \to x^*$ as $j \to \infty$, we also have $\lim_{j \to \infty} F(x^{(t_j+1)}) = F(x^*) = F^*$. Then, from Eqs. (49), (50), and the continuity of $F$, it is easy to see that $\|x^{(t+1)} - x^*\|_2^2 \leq \|v^{(t+1)} - x^*\|_2^2$, which leads to

$$\|x^{(t)} - P(x^{(t)})\|_2^2 \leq \left(\frac{\beta}{1-\beta}\right)\left[\|x^{(t)} - x^*\|_2^2 - \|x^{(t+1)} - x^*\|_2^2\right]. \tag{52}$$

By averaging this inequality over $T$ iterations and dropping the last term $\|x^{(t+1)} - x^*\|_2^2$, we obtain

$$\frac{1}{T}\sum_{t=1}^{T}\|x^{(t)} - P(x^{(t)})\|_2^2 \leq \frac{2}{T}\left(\frac{1+\kappa}{1-\kappa}\right)\|x^{(0)} - x^*\|_2^2. \tag{53}$$

To obtain the result that depends on $\kappa \in (0,1)$, we note that for any $\alpha \in (0, 1/L]$, we write

$$\frac{\beta}{1-\beta} = \frac{\kappa + \alpha L/2 - \kappa\alpha L}{(1-\kappa)(1-\alpha L/2)} \leq \frac{\kappa + \frac{1}{2}}{\frac{1-\kappa}{2}} \leq 2\left(\frac{1+\kappa}{1-\kappa}\right). \tag{54}$$

Thus, from Eqs. (53) and (54) the result holds. $\square$

## 7.1 Pseudo-code for plug-and-play invex imaging

For the sake of completeness we present Algorithm 3 which is the pseudo-code of the plug-and-play version of APG for solving Eq. (12). The scaled-up convergence of APG are offered by two auxiliary variables, i.e., $y^{(t+1)}$ and $z^{(t+1)}$ in Lines 4 and 5. In Line 6 is presented the replacement of the proximal operator in APG pseudo-code with a neural network based denoiser Noise2Void [11]. And a monitor constrain computed in Line 8, to satisfy the sufficient descent property.

---

**Algorithm 3** Plug-and-play Proximal Gradient Algorithm

1: **input**: Tolerance constant $\epsilon \in (0,1)$, initial point $x^{(0)}$, and number of iterations $T$
2: **initialize**: $x^{(1)} = x^{(0)} = z^{(0)}, r_1 = 1, r_0 = 0, \alpha_1, \alpha_2 < \frac{1}{L}$, and $\lambda \in (0,1]$
3: **for** $t = 1$ to $T$ **do**
4:     $y^{(t)} = x^{(t)} + \frac{r_{t-1}}{r_t}(z^{(t)} - x^{(t)}) + \frac{r_{t-1}-1}{r_t}(x^{(t)} - x^{(t-1)})$
5:     $z^{(t+1)} = \text{prox}_{\alpha_2\lambda g}(y^{(t)} - \alpha_2\nabla f(y^{(t)}))$
6:     $v^{(t+1)} = \text{Noise2Void}(x^{(t)} - \alpha_1\nabla f(x^{(t)}))$     $\triangleright$ This calls the trained Noise2Void model
7:     $r_{t+1} = \frac{\sqrt{4(r_t)^2+1}+1}{2}$
8:     $x^{(t+1)} = \begin{cases} z^{(t+1)}, & \text{if } f(z^{(t+1)}) + \lambda g(z^{(t+1)}) \leq f(v^{(t+1)}) + \lambda g(v^{(t+1)}) \\ v^{(t+1)}, & \text{otherwise} \end{cases}$
9: **end for**
10: **return**: $x^{(T)}$

---

# 8 Proof of Lemma 5

*Proof.* To prove this lemma we start exploiting the convexity of $\lambda g(\boldsymbol{x}) + \frac{1}{2}\|\boldsymbol{x} - \boldsymbol{u}\|_2^2$ for fixed $\boldsymbol{v} \in \mathbb{R}^n$ according to Theorem 3, and $\lambda \in (0, 1]$. Then, for all functions in Table 2, we have

$$\lambda g(\boldsymbol{x}) + \frac{1}{2}\|\boldsymbol{x} - \boldsymbol{u}\|_2^2 - \lambda g(\boldsymbol{y}) - \frac{1}{2}\|\boldsymbol{y} - \boldsymbol{u}\|_2^2 \geq \left(\boldsymbol{\zeta}_y + \boldsymbol{y} - \boldsymbol{u}\right)^T (\boldsymbol{x} - \boldsymbol{y})$$

$$\lambda g(\boldsymbol{x}) - \lambda g(\boldsymbol{y}) \geq \left(\boldsymbol{\zeta}_y + \boldsymbol{y} - \boldsymbol{u}\right)^T (\boldsymbol{x} - \boldsymbol{y}) + \frac{1}{2}\|\boldsymbol{y} - \boldsymbol{u}\|_2^2 - \frac{1}{2}\|\boldsymbol{x} - \boldsymbol{u}\|_2^2$$

$$\lambda g(\boldsymbol{x}) - \lambda g(\boldsymbol{y}) \geq \left(\boldsymbol{\zeta}_y + \boldsymbol{y} - \boldsymbol{u}\right)^T (\boldsymbol{x} - \boldsymbol{y}) + (\boldsymbol{x} - \boldsymbol{u})^T (\boldsymbol{y} - \boldsymbol{x}) \tag{55}$$

for all $\boldsymbol{x}, \boldsymbol{y} \in \mathbb{R}^n$, and $\boldsymbol{\zeta}_y \in \partial \lambda g(\boldsymbol{y})$, where the third inequality comes from the convexity of $f(\boldsymbol{x}) = \frac{1}{2}\|\boldsymbol{x} - \boldsymbol{u}\|_2^2$. Then, from Eq. (55) we conclude

$$\lambda g(\boldsymbol{x}) - \lambda g(\boldsymbol{y}) \geq \boldsymbol{\zeta}_y^T (\boldsymbol{x} - \boldsymbol{y}) - \|\boldsymbol{x} - \boldsymbol{y}\|_2^2, \tag{56}$$

for all $\boldsymbol{x}, \boldsymbol{y} \in \mathbb{R}^n$, and $\boldsymbol{\zeta}_y \in \partial \lambda g(\boldsymbol{y})$.

The iterative procedure summarized in Algorithm 2 is seen as

$$\boldsymbol{x}^{(t+1)} = \arg\min_{\boldsymbol{x} \in \mathbb{R}^n} \ \left\langle \nabla f(\boldsymbol{x}^{(t)}), \boldsymbol{x} - \boldsymbol{x}^{(t)} \right\rangle + \frac{1}{2\alpha_t \lambda}\|\boldsymbol{x} - \boldsymbol{x}^{(t)}\|_2^2 + g(\boldsymbol{x}) \tag{57}$$

We write equal in the above equation because the proximal in Eq. (11) is invex therefore it always map to a global optimizer. From the Lipschitz continuous of $\nabla f$ we have

$$f(\boldsymbol{x}^{(t+1)}) \leq f(\boldsymbol{x}^{(t)}) + \left\langle \nabla f(\boldsymbol{x}^{(t)}), \boldsymbol{x}^{(t+1)} - \boldsymbol{x}^{(t)} \right\rangle + \frac{L}{2}\|\boldsymbol{x}^{(t+1)} - \boldsymbol{x}^{(t)}\|_2^2. \tag{58}$$

Considering the fact that from Eq. (57) we conclude $-\nabla f(\boldsymbol{x}^{(t)}) + \frac{1}{\alpha_t \lambda}\left(\boldsymbol{x}^{(t)} - \boldsymbol{x}^{(t+1)}\right) \in \partial g(\boldsymbol{x}^{(t+1)})$, then Eq. (56) leads to

$$\lambda g(\boldsymbol{x}^{(t)}) - \lambda g(\boldsymbol{x}^{(t+1)}) \geq \left\langle -\nabla f(\boldsymbol{x}^{(t)}) + \frac{1}{\alpha_t \lambda}\left(\boldsymbol{x}^{(t)} - \boldsymbol{x}^{(t+1)}\right), \boldsymbol{x}^{(t)} - \boldsymbol{x}^{(t+1)} \right\rangle - \|\boldsymbol{x}^{(t)} - \boldsymbol{x}^{(t+1)}\|_2^2$$

$$\geq \left\langle \nabla f(\boldsymbol{x}^{(t)}), \boldsymbol{x}^{(t+1)} - \boldsymbol{x}^{(t)} \right\rangle + \left(\frac{1}{\alpha_t \lambda} - 1\right)\|\boldsymbol{x}^{(t)} - \boldsymbol{x}^{(t+1)}\|_2^2. \tag{59}$$

The above Eq. (59) combined with Eq. (58) yields

$$\lambda g(\boldsymbol{x}^{(t)}) - \lambda g(\boldsymbol{x}^{(t+1)}) \geq f(\boldsymbol{x}^{(t+1)}) - f(\boldsymbol{x}^{(t)}) - \frac{L}{2}\|\boldsymbol{x}^{(t+1)} - \boldsymbol{x}^{(t)}\|_2^2 + \left(\frac{1}{\alpha_t \lambda} - 1\right)\|\boldsymbol{x}^{(t)} - \boldsymbol{x}^{(t+1)}\|_2^2$$

$$f(\boldsymbol{x}^{(t)}) + \lambda g(\boldsymbol{x}^{(t)}) \geq f(\boldsymbol{x}^{(t+1)}) + \lambda g(\boldsymbol{x}^{(t+1)}) + \left(\frac{1}{\alpha_t \lambda} - 1 - \frac{L}{2}\right)\|\boldsymbol{x}^{(t)} - \boldsymbol{x}^{(t+1)}\|_2^2. \tag{60}$$

Observe that by taking $\alpha_t < \frac{2}{L+2}$, then from Eq. (60) we have that $f(\boldsymbol{x}^{(t)}) + \lambda g(\boldsymbol{x}^{(t)}) \geq f(\boldsymbol{x}^{(t+1)}) + \lambda g(\boldsymbol{x}^{(t+1)})$ which is a sufficient decreasing condition. In addition, considering that the list of invex functions in Table 2, and $f(\boldsymbol{x}) = \|\boldsymbol{H}\boldsymbol{x} - \boldsymbol{v}\|_2^2$ are positive, then the loss function in Eq. (12) is bounded below. Thus, in particular $f(\boldsymbol{x}^{(t)}) + \lambda g(\boldsymbol{x}^{(t)}) - (f(\boldsymbol{x}^{(t+1)}) + \lambda g(\boldsymbol{x}^{(t+1)})) \to 0$ as $t \to \infty$, which, combined with Eq. (60), implies that $\|\boldsymbol{x}^{(t)} - \boldsymbol{x}^{(t+1)}\|_2^2 \to 0$ as $t \to \infty$. The later convergence implies the existence of fixed points to the proximal iteration in Algorithm 2 (equivalently to Eq. (57)), sufficient condition to guarantee that the sequence $\{\boldsymbol{x}^{(t)}\}$ convergences to a stationary point of $f(\boldsymbol{x}) + \lambda g(\boldsymbol{x})$. Thus, since in Theorem 4 we proved the loss function in Eq. (12) is invex then $\{\boldsymbol{x}^{(t)}\}$ converges to a global minimizer. $\square$

# 9  Image Compressive Sensing Experiments Evaluated with SSIM metric

In this section we complement results of Experiments 1,2, and 3 of Section 5. We assess the imaging quality for these experiments using the structural similarity index measure (SSIM). The best and least efficient among invex functions is highlighted in boldface and underscore, respectively.

**Experiment 1** studies the effect of different invex regularizers under Algorithm 1. The numerical results of this study are summarized in Table 3. Also, we present Figure 4 which illustrates reconstructed images, for $SNR = 30$dB, obtained by Eqs. (6), (7), (8), (9), and (10), which are compared with the outputs from FISTA, TVAL3, and ReconNet. In addition, to numerically evaluate their performance we estimate the PSNR for each image.

Table 3: Comparison between convex and invex regularizers, in terms of SSIM, under Algorithm 1, using $p = 0.5$ for Eq. (6).

| SNR | (Experiment 1) Algorithm 1, $p = 0.5$ for Eq. (6). | | | | | FISTA [12] | TVAL3 [96] | ReconNet [97] |
| | Eq. (6) | Eq. (7) | Eq. (8) | Eq. (9) | Eq. (10) | $\ell_1$-norm | | |
| --- | --- | --- | --- | --- | --- | --- | --- | --- |
| $\infty$ | **0.9486** | 0.9370 | 0.9408 | 0.9332 | 0.9447 | 0.9257 | 0.9294 | 0.9220 |
| 20dB | **0.8675** | 0.8495 | 0.8554 | 0.8437 | 0.8614 | 0.8323 | 0.8380 | 0.8267 |
| 30dB | **0.9055** | 0.8944 | 0.8981 | 0.8908 | 0.9018 | 0.8836 | 0.8872 | 0.8801 |

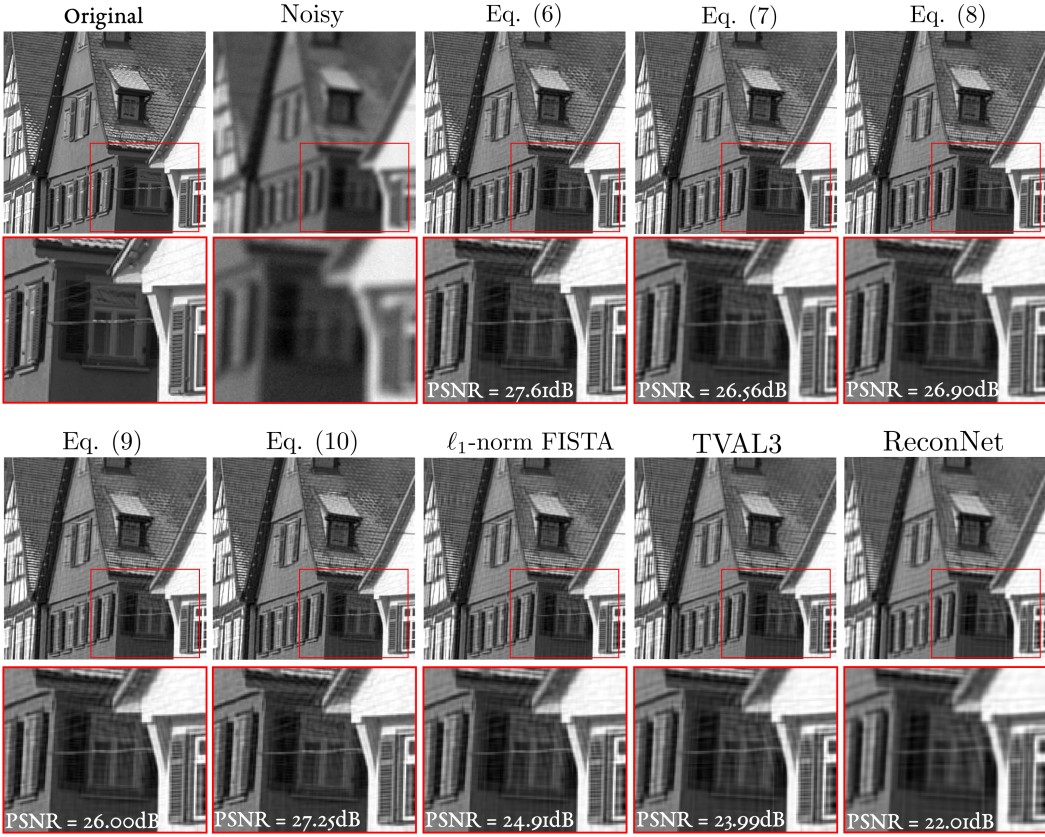

Figure 4: Reconstructed images, for $SNR = 30$dB, obtained by Algorithm 1 using Eqs. (6), (7), (8), (9), and (10), which are compared with the outputs from FISTA, TVAL3, and ReconNet. In addition, to numerically evaluate their performance we estimate the PSNR for each image.

**Experiment 2** studies the invex regularizers under the plug-and-play modification of Algorithm 1 as described in Section 4.2.2 [11]. The same deconvolution problem as in Experiment 1 is used. The numerical results of this study are summarized in Table 4. Also, we present Figure 5 which illustrates reconstructed images obtained by Eqs. (6), (7), (8), (9), and (10), which are compared with

the outputs from $\ell_1$-norm. In addition, to numerically evaluate their performance we estimate the PSNR for each image.

Table 4: Comparison between convex and invex regularizers, in terms of SSIM, under plug-and-play Algorithm 3, using $p = 0.8$ for Eq. (6).

| SNR | (Experiment 2) Algorithm 3, $p = 0.8$ for Eq. (6). | | | | | $\ell_1$-norm |
|---|---|---|---|---|---|---|
| | Eq. (6) | Eq. (7) | Eq. (8) | Eq. (9) | Eq. (10) | |
| $\infty$ | **0.9581** | 0.9409 | 0.9465 | 0.9352 | 0.9523 | 0.9297 |
| 20dB | **0.8808** | 0.8680 | 0.8722 | 0.8638 | 0.8765 | 0.8597 |
| 30dB | **0.9189** | 0.9043 | 0.9091 | 0.8995 | 0.9140 | 0.8948 |

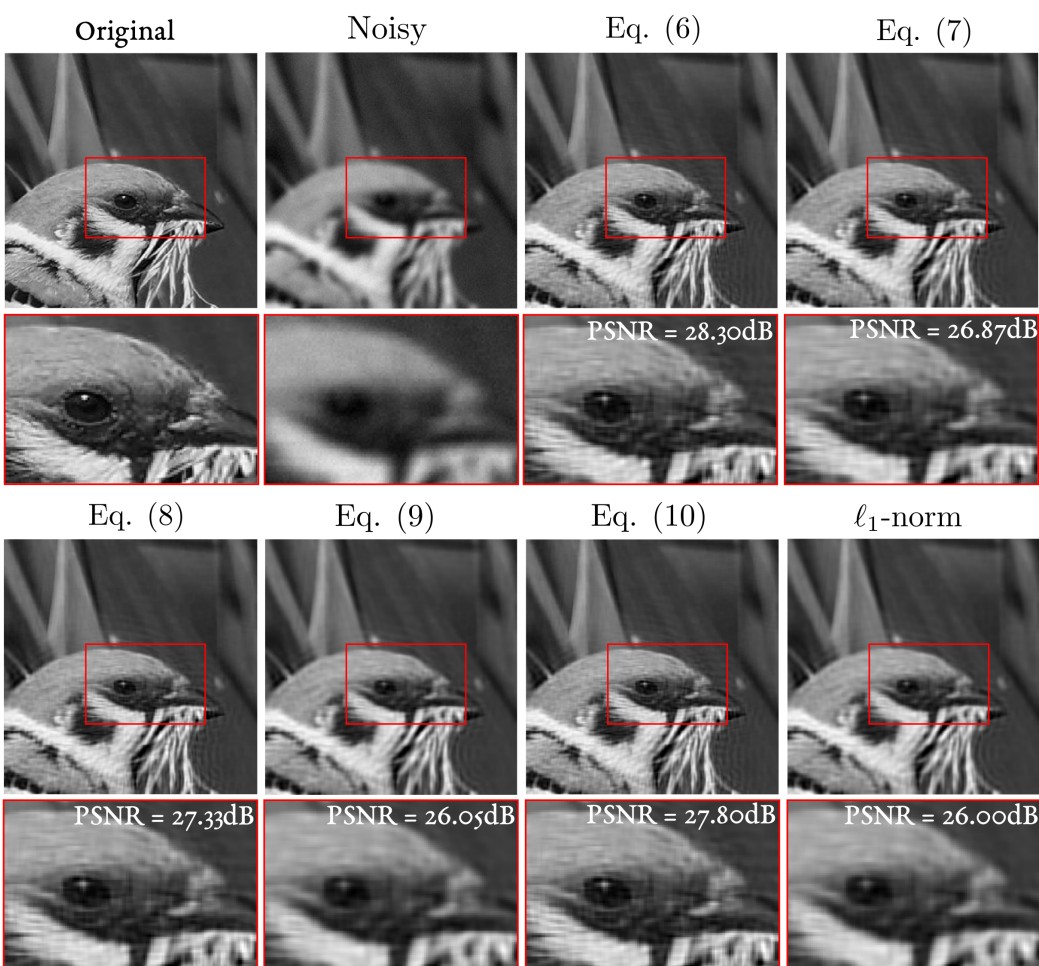

Figure 5: Reconstructed images, for $SNR = 30$dB, obtained by Algorithm 3 using Eqs. (6), (7), (8), (9), and (10), which are compared with the outputs from $\ell_1$-norm. In addition, to numerically evaluate their performance we estimate the PSNR for each image.

**Experiment 3** compares the invex regularizers but under the unrolling framework as described in Section 4.2.3. The numerical results of this study are summarized in Table 5. Also, we present Figure 6 which illustrates reconstructed images obtained by Eqs. (6), (7), (8), (9), and (10) with ISTA-Net, which are compared with the outputs from $\ell_1$-norm + ISTA-Net. In addition, to numerically evaluate their performance we estimate the PSNR for each image.

Table 5: Performance comparison between convex and invex regularizers, in terms of SSIM, for the unrolling experiment, using $p = 0.85$ for Eq. (6).

| SNR | $m/n$ | (Experiment 3) Algorithm 2 - unfolded LISTA. $p = 0.85$ for Eq. (6) | | | | | LISTA [13] | ReconNet [97] |
| --- | --- | --- | --- | --- | --- | --- | --- | --- |
| | | Eq. (6) | Eq. (7) | Eq. (8) | Eq. (9) | Eq. (10) | $\ell_1$-norm | |
| | 0.2 | **0.9279** | 0.9132 | 0.9181 | 0.9084 | 0.9230 | 0.9037 | 0.8990 |
| $\infty$ | 0.4 | **0.9610** | 0.9423 | 0.9485 | 0.9363 | 0.9547 | 0.9303 | 0.9244 |
| | 0.6 | **0.9890** | 0.9620 | 0.9708 | 0.9533 | 0.9798 | 0.9448 | 0.9364 |
| | 0.2 | **0.8690** | 0.8628 | 0.8649 | 0.8608 | 0.8669 | 0.8587 | 0.8567 |
| 20dB | 0.4 | **0.9370** | 0.9205 | 0.9259 | 0.9151 | 0.9314 | 0.9098 | 0.9045 |
| | 0.6 | **0.9498** | 0.9411 | 0.9440 | 0.9382 | 0.9469 | 0.9353 | 0.9325 |
| | 0.2 | **0.8876** | 0.8781 | 0.8812 | 0.8750 | 0.8844 | 0.8719 | 0.8688 |
| 30dB | 0.4 | **0.9510** | 0.9318 | 0.9381 | 0.9255 | 0.9445 | 0.9194 | 0.9133 |
| | 0.6 | **0.9619** | 0.9545 | 0.9569 | 0.9520 | 0.9594 | 0.9496 | 0.9472 |
| | | (Experiment 3) Algorithm 2 - unfolded ISTA-Net. $p = 0.85$ for Eq. (6) | | | | | | |
| SNR | $m/n$ | Eq. (6) | Eq. (7) | Eq. (8) | Eq. (9) | Eq. (10) | $\ell_1$-norm [98] | |
| | 0.2 | **0.9350** | 0.9219 | 0.9262 | 0.9176 | 0.9306 | 0.9134 | |
| $\infty$ | 0.4 | **0.9733** | 0.9541 | 0.9604 | 0.9479 | 0.9668 | 0.9417 | |
| | 0.6 | **0.9899** | 0.9697 | 0.9763 | 0.9632 | 0.9831 | 0.9567 | |
| | 0.2 | **0.8829** | 0.8745 | 0.8773 | 0.8717 | 0.8801 | 0.8690 | |
| 20dB | 0.4 | **0.9501** | 0.9323 | 0.9382 | 0.9265 | 0.9441 | 0.9208 | - |
| | 0.6 | **0.9611** | 0.9520 | 0.9550 | 0.9490 | 0.9580 | 0.9460 | |
| | 0.2 | **0.8990** | 0.8836 | 0.8887 | 0.8786 | 0.8938 | 0.8736 | |
| 30dB | 0.4 | **0.9641** | 0.9437 | 0.9504 | 0.9370 | 0.9572 | 0.9305 | |
| | 0.6 | **0.9859** | 0.9695 | 0.9749 | 0.9641 | 0.9804 | 0.9588 | |

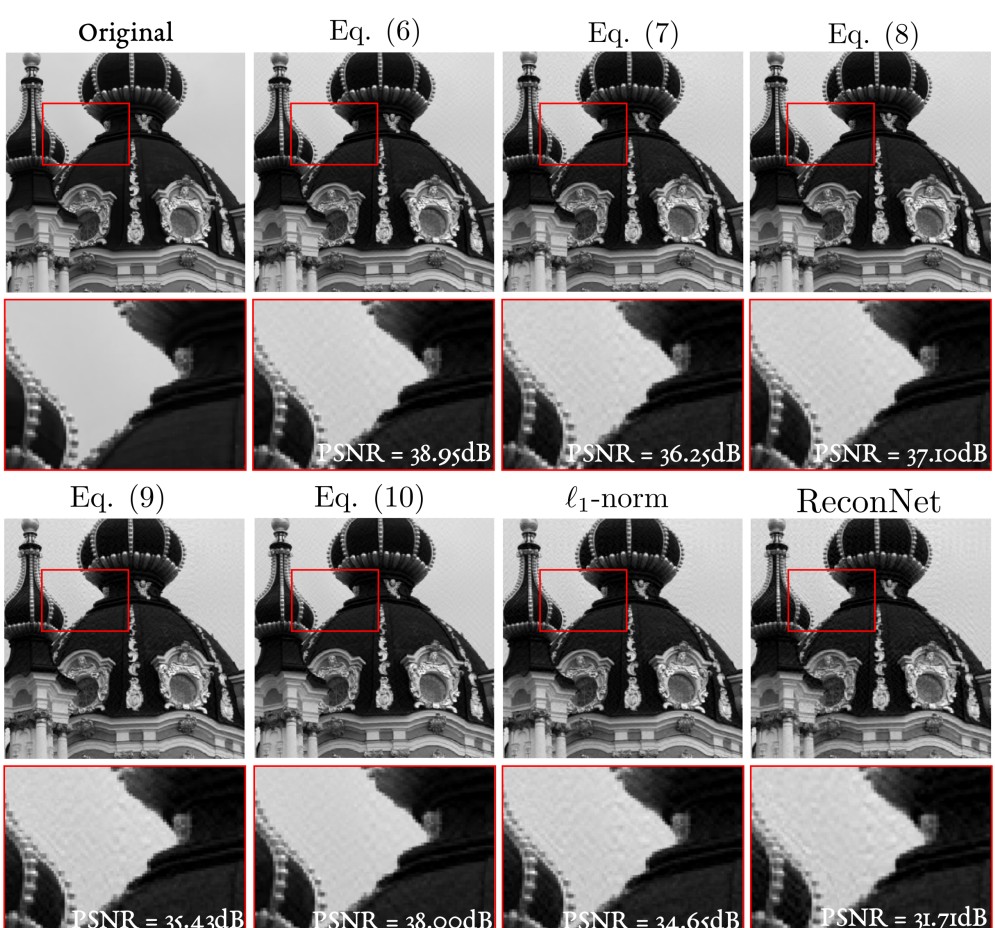

Figure 6: Reconstructed images, for $SNR = 30$dB, obtained by ISTA-Net using Eqs. (6), (7), (8), (9), and (10), which are compared with the outputs from $\ell_1$-norm, where $m/n = 0.6$. In addition, to numerically evaluate their performance we estimate the PSNR for each image.

## 10  Image Denoising Illustration

For the sake of completeness we present in Algorithm 4 the denoising procedure employed in this paper (following [14]) using invex regularizers $g(\boldsymbol{x})$ in Eqs. (6), (8), and (10). The parameters $\lambda_1, \lambda_2, K$, and $S$ were chosen to be the best for each analyzed function determined by cross validation.

---

**Algorithm 4** Denoising procedure using invex regularizers

---

1: **input**: noisy image $\boldsymbol{x}$, $S$ the number of patches of size $16 \times 16$, $K$ number of iterations, and constant $\lambda_1, \lambda_2 \in (0, 1]$.
2: **initialize**: $\boldsymbol{W}^{(0)} = \frac{1}{256}\mathbf{1}$ where $\mathbf{1} \in \mathbb{R}^{256 \times 256}$ is the matrix of ones.
3: **Compute:** $\boldsymbol{P} \in \mathbb{R}^{256 \times S}$ matrix containing random patches of size $16 \times 16$ from $\boldsymbol{x}$
4: $\boldsymbol{A} = (\boldsymbol{I}_{256} - \boldsymbol{W}^{(0)}(\boldsymbol{W}^{(0)})^T)\boldsymbol{P}$, where $\boldsymbol{I}_{256} \in \mathbb{R}^{256 \times 256}$ is the identity matrix
5: **for** $t = 1$ to $K$ **do**
6: $\qquad \boldsymbol{W}^{(t)} = (\boldsymbol{W}^{(t-1)})^T \boldsymbol{P}$
7: $\qquad \hat{\boldsymbol{W}}^{(t)}[i,j] = \begin{cases} \boldsymbol{W}[i,j] & |\boldsymbol{W}[i,j]| \leq \lambda_1 \\ 0 & \text{otherwise} \end{cases}$
8: $\qquad$ run the SVD decomposition on $\boldsymbol{A}(\hat{\boldsymbol{W}}^{(t)})^T$ such that $\boldsymbol{A}(\hat{\boldsymbol{W}}^{(t)})^T = \boldsymbol{U}\boldsymbol{D}\boldsymbol{V}^T$.
9: $\qquad \boldsymbol{W}^{(t)} = \boldsymbol{U}\boldsymbol{V}^T$
10: **end for**
11: $\hat{\boldsymbol{x}} = (\boldsymbol{W}^{(K)})^T \text{Prox}_{\lambda_2 g}(\boldsymbol{W}^{(K)}\boldsymbol{x})$ $\qquad\qquad\qquad\qquad\qquad$ ▷ Denoising step
12: **return:** $\hat{\boldsymbol{x}}$ $\qquad\qquad\qquad\qquad\qquad\qquad\qquad\qquad\qquad\quad$ ▷ Denoised image

---

Employing Algorithm 4, in Figure 7 we present some denoised images obtained by Eqs. (6), (8), (10), which are compared with the outputs from BM3D, and Noise2Void. Since we are analyzing all the regularizers under non-ideal scenarios due to noise, results in Figure 7 highlight the benefit of having invex regularizers since the cleanest image is obtained by Eq. (6). In addition, to numerically evaluate their performance we employ the structural similarity index measure (SSIM) by reporting the SSIM map for each denoised image and its averaged value. Recall that SSIM is reported in the range $[0, 1]$ where 1 is the best achievable quality and 0 the worst. In the SSIM map small values of SSIM appear as dark pixels. Thus, we conclude the best performance is achieved using the regularizer in Equation (6) since it has the whitest SSIM maps (with highest SSIM values).

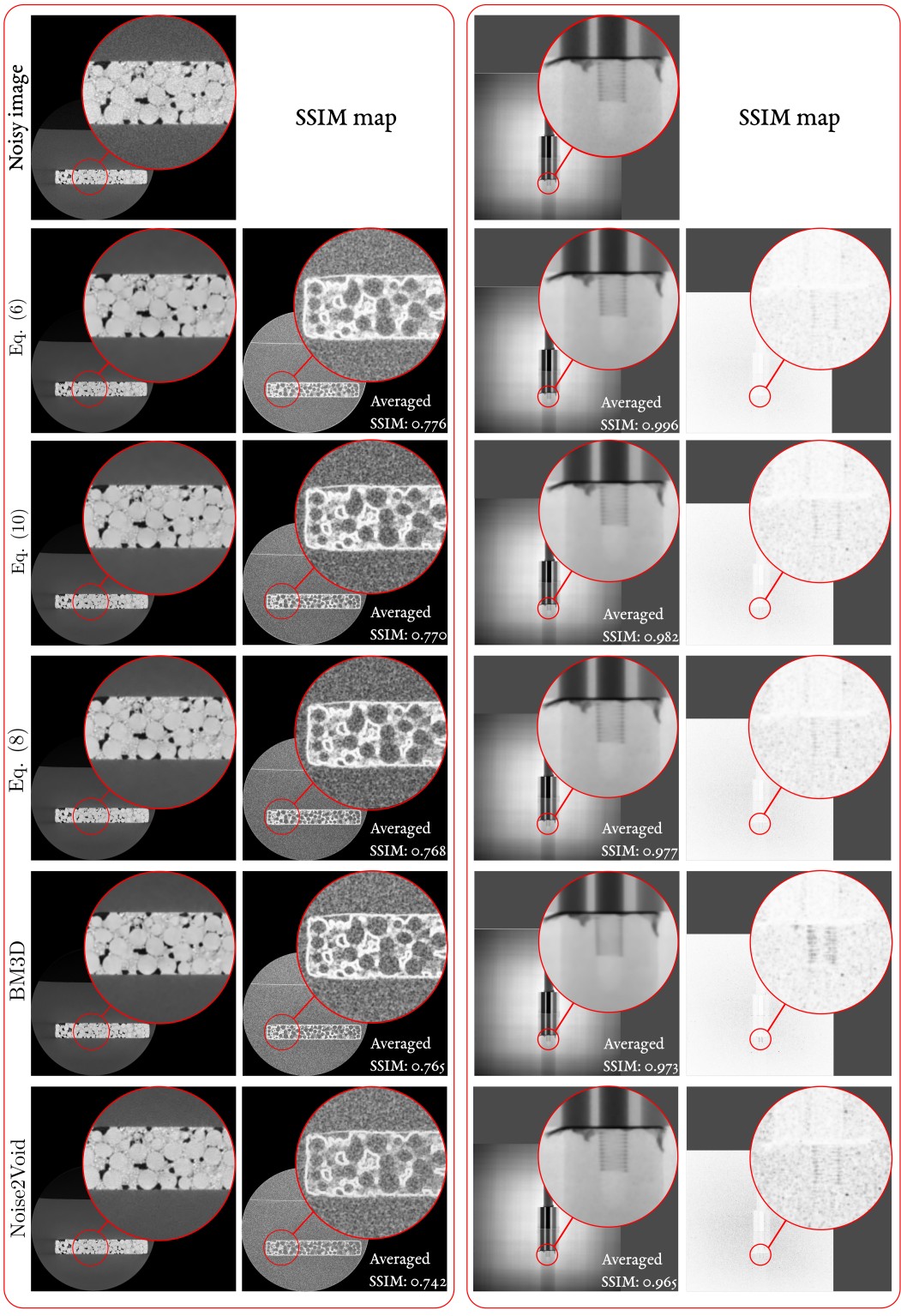

Figure 7: Denoised image illustration for Eqs. (6),(8), (10) and the state-of the-arts BM3D and Noise2Void. To evaluate the performance we employ the structural similarity index measure (SSIM) by reporting the SSIM map for each denoised image and its averaged value. Recall that SSIM is reported in the range $[0, 1]$ where 1 is the best achievable quality and 0 the worst. In the SSIM map small values of SSIM appear as dark pixels. Thus, we conclude the best performance is achieved using the regularizer in Eq. (6) since it has the whitest SSIM maps.