# OpenReview forum: "Improved Imaging by Invex Regularizers with Global Optima Guarantees"
_NeurIPS.cc/2022/Conference — NeurIPS 2022 Accept_

### Official Review · Reviewer_xZpv · 2022-07-11

**Rating:** 7
**Confidence:** 3
**Soundness:** 3 good
**Presentation:** 3 good
**Contribution:** 3 good

**Summary:**

This manuscript revisits the concept of invexity and provides the first list of proved invex regularizers for improving image reconstruction. Convergence conditions are analyzed and experiments are conducted on practical applications.

**Questions:**

1.	Can you compare the performance with some CNN-based baselines?
2.	For different regularizers, were the parameters (e.g. \lambda in eq.12) chosen as the same value?


**Limitations:**

No potential negative societal impact is mentioned. Both theoretical analysis and practical applications of invex functions are provided. I believe the proposed finite list will inspire exploration in related fields.

**Strengths And Weaknesses:**

Strength:
1.	The paper is well-written with compact structure and clear interpretations.
2.	Some useful regularizers in imaging applications are formally studied in a new perspective of invex function. Besides, a simple and valid way of constructing an invex function is provided, which may potentially inspire later researchers.
3.	Experimental comparisons indicate that the proposed invex regularizers have potential to improve performance in practical applications.
Weakness:
The proposed new invex function doesn’t achieve the best efficiency compared with the existing non-convex functions.

---

> ### Author Response · Authors · 2022-08-02
> **Author Response for Official Review by Reviewer xZpv**
>
> We thank you for positive comments and the accurate summary of our work. We address your insightful comments as follows.
>
> # Questions
> * **Q1.** We have addressed this question by including two CNN-based baselines, namely, ReconNet and ISTA-net. We have included the relevant results in the expanded Table 3 (main paper) and in the newly added Tables 5, and 7 (Appendix I of the supplemental material), in the revised draft. Please see below the mentioned Tables.
> ---
> |Table 3. Performance comparison, in terms of PSNR (dB)|
> ---
>
> |(Experiment 1) Algorithm 1, p = 0.5 for Eq(6)|
> ---
>
> |SNR|Eq(6)|Eq(7)|Eq(8)|Eq(9)|Eq(10)|FISTA[13]|ReconNet[97]|TVAL3[96]|SCAD[94]|MCP[24]|
> |---|---|---|---|---|---|---|---|---|---|---|
> |$\infty$|**33.40**|31.25|30.00|32.65|32.65|29.97|27.01|28.77|30.55|31.30|
> |20dB|**24.60**|22.83|23.39|22.00|23.98|21.80|19.99|20.49|22.60|23.01|
> |30dB|**27.61**|26.56|26.90|26.00|27.25|24.91|22.01|23.99|26.10|26.77|
> ---
>
> |(Experiment 3) Algorithm 2 - unfolded LISTA. $p = 0.85$ for Eq(6)|
> ---
>
> |SNR|$m/n$|Eq(6)|Eq(7)|Eq(8)|Eq(9)|Eq(10)|$\ell_{1}$-norm[86]|ReconNet[97]|
> |---|---|---|---|---|---|---|---|---|
> |$\infty$|0.2|**31.32**|29.20|29.87|28.56|30.58|27.95|26.59|
> ||0.4|**36.10**|33.50|34.34|32.75|35.20|32.01|31.86|
> ||0.6|**41.27**|37.81|38.90|36.09|40.05|35.82|34.42|
> |20dB|0.2|**26.00**|24.45|24.94|23.97|25.01|23.52|22.00|
> ||0.4|**32.67**|30.64|31.32|30.02|32.29|29.43|28.24|
> ||0.6|**34.38**|33.00|33.28|32.94|33.64|32.60|30.20|
> |30dB|0.2|**27.65**|26.20|26.66|25.75|27.15|25.32|23.64|
> ||0.4|**34.33**|31.89|32.66|31.02|33.47|30.46|29.88|
> ||0.6|**37.03**|34.84|35.54|34.17|36.27|33.53|31.71|
> ---
>
> |(Experiment 3) Algorithm 2 - unfolded ISTA-Net. $p = 0.85$ for Eq(6)|
> ---
>
> |SNR|$m/n$|Eq(6)|Eq(7)|Eq(8)|Eq(9)|Eq(10)|$\ell_{1}$-norm[98]|
> |---|---|---|---|---|---|---|---|
> |$\infty$|0.2|**32.50**|30.15|30.89|29.04|31.67|28.77|
> ||0.4|**38.33**|35.72|36.55|34.92|37.41|34.17|
> ||0.6|**43.61**|40.07|41.18|39.02|42.36|38.02|
> |20dB|0.2|**28.29**|26.22|26.87|25.60|27.56|25.01|
> ||0.4|**33.96**|32.11|32.71|31.55|33.32|31.00|
> ||0.6|**35.77**|34.68|35.03|34.33|35.39|33.99|
> |30dB|0.2|**29.34**|28.30|28.63|27.97|28.98|27.65|
> ||0.4|**35.41**|33.33|33.99|32.69|34.68|32.08|
> ||0.6|**38.95**|36.25|37.10|35.43|38.00|34.65|
> ---
>
> ---
> |Table 5. Performance comparison, in terms of SSIM. (Experiment 1) Algorithm 1, $p = 0.5$ for Eq(6)|
> ---
>
> |SNR|Eq(6)|Eq(7)|Eq(8)|Eq(9)|Eq(10)|FISTA[13]|TVAL3[96]|ReconNet[97]|
> |---|---|---|---|---|---|---|---|---|
> |$\infty$|**0.9486**|0.937|0.940|0.933|0.944|0.925|0.929|0.922|
> |20dB|**0.8675**|0.849|0.855|0.843|0.861|0.832|0.838|0.826|
> |30dB|**0.9055**|0.894|0.898|0.890|0.901|0.883|0.887|0.880|
> ---
>
> |Table 7. Performance comparison, in terms of SSIM|
> ---
>
> |(Experiment 3) Algorithm 2 - unfolded LISTA. $p = 0.85$ for Eq(6)|
> ---
>
> |SNR|$m/n$|Eq(6)|Eq(7)|Eq(8)|Eq(9)|Eq(10)|$\ell_{1}$-norm[86]|ReconNet[97]|
> |---|---|---|---|---|---|---|---|---|
> |$\infty$|0.2|**0.927**|0.913|0.918|0.908|0.923|0.903|0.899|
> ||0.4|**0.961**|0.942|0.948|0.936|0.954|0.930|0.924|
> ||0.6|**0.989**|0.962|0.970|0.953|0.979|0.944|0.936|
> |20dB|0.2|**0.869**|0.862|0.864|0.860|0.866|0.858|0.856|
> ||0.4|**0.937**|0.920|0.925|0.915|0.931|0.909|0.904|
> ||0.6|**0.949**|0.941|0.944|0.938|0.946|0.935|0.932|
> |30dB|0.2|**0.887**|0.878|0.881|0.875|0.884|0.871|0.868|
> ||0.4|**0.951**|0.931|0.938|0.925|0.944|0.919|0.913|
> ||0.6|**0.961**|0.954|0.956|0.952|0.959|0.949|0.947|
> ---
>
> |(Experiment 3) Algorithm 2 - unfolded ISTA-Net. p = 0.85 for Eq(6)|
> ---
>
> |SNR|$m/n$|Eq(6)|Eq(7)|Eq(8)|Eq(9)|Eq(10)|$\ell_{1}$-norm[86]|
> |---|---|---|---|---|---|---|---|
> |$\infty$|0.2|**0.935**|0.921|0.926|0.917|0.930|0.913|
> ||0.4|**0.973**|0.954|0.960|0.947|0.966|0.941|
> ||0.6|**0.990**|0.969|0.976|0.963|0.983|0.956|
> |20dB|0.2|**0.882**|0.874|0.877|0.871|0.880|0.869|
> ||0.4|**0.950**|0.932|0.938|0.926|0.944|0.920|
> ||0.6|**0.961**|0.952|0.955|0.949|0.958|0.946|
> |30dB|0.2|**0.899**|0.950|0.932|0.968|0.915|0.987|
> ||0.4|**0.964**|0.943|0.950|0.937|0.957|0.930|
> ||0.6|**0.985**|0.969|0.974|0.964|0.980|0.958|
> ---
>
> * **Q2.** The parameter value for $\lambda$ was variable. The best value for $\lambda$ was chosen using cross validation. We have stated this  in Experiments and Results section (see line 285 in the revised draft).
>
> # Discussion about weaknesses
> * **1.** We have not included efficiency or computational complexity studies as part of this work, as we considered them as out of scope. Our focus, instead, has been around potential utility of invex functions in imaging applications supported by theoretical guarantees than developing a best-performing regularizer. We will endeavour to optimize the efficiency of our regularizers as part of our future work. Thank you for pointing this out.

---

> > ### Comment · Reviewer_xZpv · 2022-08-05
> > **All my concerns have been addressed**
> >
> > Thank you for the answers to my questions. I have also read the comments of other reviewers and the replies of the authors. I think manuscript can be accept.

---

> > > ### Author Response · Authors · 2022-08-06
> > > **Thank you!**
> > >
> > > We are glad to know that the reviewer found our rebuttal helpful. Thank you for the valuable and constructive comments, which greatly helped us to improve the manuscript.

---

### Official Review · Reviewer_2EmJ · 2022-07-11

**Rating:** 7
**Confidence:** 3
**Soundness:** 4 excellent
**Presentation:** 4 excellent
**Contribution:** 4 excellent

**Summary:**

The authors consider the task of image reconstruction with regularization. Beginning at least a few decades past, regularizers have been adapted by various communities first from L2 quadratic penalties to L1-like edge-preserving penalties. More recently, motivated by compressed sensing researchers have used L1 or even non-convex penalties. Non-convex penalties are more difficult to analyze from a theoretical perspective, but empirically have been observed to produce the best results for image reconstruction tasks.

In this manuscript the authors propose to focus on classes of regularizers that are invex functions. Invex functions are functions where "at any point where the derivative vanishes (stationary point), it is a global minimizer of the function." The manuscript's contributions are three-fold: 1) the manuscript provides the first list of regularizers with proved invexity for image reconstruction, 2) the manuscript establishes convergence guarantees for three types of image reconstruction problems with invex regularizers and 3) the manuscript provides empirical data on the performance of the regularizers.

**Questions:**

1. How can the H matrix in Theorem 4 be unitary if m < n? If m < n, H cannot have an orthonormal basis for vectors of size n.
2. Could the annotation of Table 3 be improved? This is one of the most important results of the paper. I would bold the best-performing PSNR value in each row as is standard practice at machine learning conferences rather than the regularizer class. Also, the performance of Eq. (6) should be discussed.
3. How were the datasets used? Did the authors merge all datasets under consideration? These details should be further specified for the experiments.

**Limitations:**

The authors discussed limitations under the relevant theorems and lemmas, but for practitioners I think it would be better to have a dedicated Limitations section. A few items that might be considered for such a section:

- The plug-and-play result only shows that the optimization sequences converges to a close estimate of the solution.
- The theory does not seem to consider noisy measurements.
- The reconstruction tasks under consideration seem to be limited to deconvolution.

**Strengths And Weaknesses:**

**Strengths**
- The authors present convergence theory for three problems of substantial interest to the reconstruction community: denoising, compressed sensing, and plug-and-play reconstruction. Each of these has been applied in practical settings. The paper includes global convergence guarantees for each setting.
- The invex approach seems to be a new and interesting perspective into inverse problems. In particular, the quasi-norm of Equation (6) has long been of great interest to the image reconstruction community due to its relations to compressed sensing, so it is encouraging to see that the current manuscript's theory can be applied to this case.
- In addition to theoretical results, the paper also includes empirical results for each of the regularizers under consideration.

**Weaknesses**
- Some properties of H in Theorem 4 seem inconsistent with how it is used in the subsequent algorithms. For example, Theorem 4 states that H is unitary, but H is elsewhere specified to have full row rank. The specific properties of H necessary for each theorem setting should be further clarified.
- The presentation of Table 3 is somewhat confusing.
- The variations in invex regularizer performance is not discussed.

---

> ### Author Response · Authors · 2022-08-02
> **Author Response for Official Review by Reviewer 2EmJ**
>
> We thank you for positive comments. We address the comments by grouping them into three categories: questions, discussion about weaknesses and limitations. The numbering of the references are from the new version of the manuscript.
>
> # Questions
> * **Q1.** We apologise for the incorrect statement of $\boldsymbol{H}$ being unitary. We have now fixed this by revising Theorem 4, where  $\boldsymbol{H}$ should be a matrix with $\ell_{2}$-normalized columns satisfying the restricted isometry property (RIP) for $\delta_{2k}<\frac{1}{3}$ where $k<n$ is the sparsity. This is a standard assumption in compressive sensing [Theorem 5.3,70].
>
> * **Q2.** We have improved the annotations of Table 3, by highlighting the best and least efficient invex regularizers for every row. We have also restructured it to  organize better the results. We have also discussed the performance of Equation (6),  as follows: **The intuition behind the superiority of (6) comes from the possibility of adjusting the value of $p$ in data-dependent manner [49]. This means that when the images are strictly sparse, and the noise is relatively low, a small value of $p$ should be used. Conversely, when images are non-strictly sparse and/or the noise is relatively high, a larger value of $p$ tend to yield better performance (which seems to be the case for the selected image datasets).** (see line 372 in the revised draft).
>
> * **Q3.** Yes, for the numerical tests we did merged all datasets under consideration (mentioned in Section 5), and after this we created the corresponding training and testing image sets (for learning  and evaluation scenarios). We have clarified better about how the datasets are used in the evaluation,  by stating  **A number of  datasets have been merged to formulate one unique dataset for our training and evaluation purposes. These are DIV2K super-resolution[88], the McMaster[89], Kodak[90],  Berkeley Segmentation (BSDS 500) [91], Tampere Images (TID2013) [92] and the Color BSD68 [93] datasets.** (see line 272 in the revised draft).
>
> # Discussion about weaknesses
> * **1.** Thank you for pointing this out, and we agree this is incorrect. We have addressed this issue as stated above. The changes we made conform to the full-rank condition required by Lemma 2 when $\boldsymbol{H}$ satisfies RIP. This result is important to invex community, because it supports the validity of Lemma 2 to build invex functions with affine mappings. We have modified the manuscript as follows to explain this more clearly: In Section 3 (line 123 in the revised draft), **"This condition on $\boldsymbol{H}$ is a mild assumption, because we show in Section 4 imaging application examples that satisfy this criterion**.  In Section 4.2 (line 196 in the revised draft), **We clarify that if $\boldsymbol{H}$ satisfies the mentioned RIP, then each sub-matrix with $k$-columns of $\boldsymbol{H}$ selected according to indices of the non-zero elements of the $k$-sparse signal is a full row-rank matrix.**  (This supports the validity of Lemma 2 to build invex functions with affine mappings.)
>
> * **2.** We have improved the presentation of Table 3 by annotating the results better, by improving its organization, and by adding additional discussions.
>
> * **3.** We have addressed this issue by including a discussion in Section 6 as follows:  **We believe that the remaining invex, SCAD, and MCP regularizers have a lesser performance than Eq. (6) as they do not have the flexibility of adjustment to the sparsity of the data. In fact, Eq. (9) shows the poorest performance because in the proof of Theorem 3, we theoretically guarantee that Eq. (9) cannot  sparsify all images. Therefore, this analysis leads to the conclusion that the invex function Eq. (6) offers the best performance for the metrics concerns and the imaging problems studied here.**
>
> # Limitations
> * **1.** We thank you and agree with the comment. Given the scarcity of the space, we have included a discussion around limitations as part of Section 6 (instead of a dedicated section), and changed the section heading to reflect this ("Discussion, Limitations and Conclusions"). The text reads as follows: **Although we have presented theoretical results with global optima using invexity, these are not without limitations. To state a few, first, our study here was focused on the reconstruction of the images under ideal conditions, such as noise-free cases (Theorem 4). However, real-world datasets will have complex noise models. Secondly, the numerical results here are focused on denoising, and deconvolution tasks where the convergence guarantees for the plug-and-play result only ensures a close estimate of the solution (Lemma 4). Although this was useful, it is desirable to explore avenues for improving convergence guarantees to global optima. Addressing these concerns will form the basis for our future work which can radically improve downstream tasks like invex robust image reconstruction, and deep learning research.**

---

> > ### Comment · Reviewer_2EmJ · 2022-08-05
> > **Reviewer 2EmJ response to authors**
> >
> > Thank you for answering all my questions. All of my comments have been addressed and I maintain my recommendation to accept.

---

> > > ### Author Response · Authors · 2022-08-06
> > > **Thank you!**
> > >
> > > Thank you for your helpful comments and support.

---

### Official Review · Reviewer_8pbb · 2022-07-13

**Rating:** 8
**Confidence:** 4
**Soundness:** 3 good
**Presentation:** 2 fair
**Contribution:** 4 excellent

**Summary:**

The paper provides convergence results attached with invex regularizers. Few non-convex regularizers with no global convergence guarantees profited from this analysis through invexity. Authors establish convergence guarantees to global optima for various advanced image reconstruction techniques with invex regularizers.  Authors prove few functions as invex functions and provide the proximal operators corresponding to them. These invex regularizers were utilized for various imaging tasks.


**Questions:**

Suggestions: (I am not asking authors to work on this right away. These are merely suggestions for a camera-ready version.)
1. The paper presentation could be improved. A comparative plot between all invex functions would've clarified things.
2. A visual aid with optimization landscape of Eq (11) i.e., $\| x - u \|^2$ + Invex function, will give more insights into the convexity of the problem, and comparison with $\| x - u \|^2 + \ell_1$ will help us understand why invex regularizers are performing better than convex regularizers.
3. Convergence analysis has been provided in theory. It would be better if the comparison of the algorithms has been provided in practice (experimental validation). plots with reconstruction MSE vs iterations, MSE vs time would be suggestible.
4. Proximal operators of non-convex penalties are known to have longer computation time. So, a comparison of the proposed methods running times will establish solid ground for the paper.

**Limitations:**

The authors discussed the limitations of their work.

**Strengths And Weaknesses:**

Strengths:
1. Invex regularizers give a decent theoretical analysis of a few non-convex regularizers.
2. Experiments included various image restoration tasks such as compressed image recovery and image denoising.

Acknowledgments:
1. Authors included more comparisons with TVAL3, ReconNet, and ISTA-Net as part of the revision.
2. More Image reconstruction examples are provided in the supplementary document as part of the revision.

Issues:
1. The revised paper is more than ten pages long. I believe it has to be limited to 9 pages until acceptance.

---

> ### Author Response · Authors · 2022-08-02
> **Author Response for Official Review by Reviewer 8pbb**
>
> We address the comments by grouping them into two categories: questions and discussion about weaknesses. The numbering of the references are from the new version of the manuscript.
> # Questions
> * **Q1.** We respectfully point out it is incorrect to state that $h(\boldsymbol{x})$ is always non-convex when we sum the $\ell_{2}$-norm and a non-convex $g(\boldsymbol{x})$. We state this following Theorem 3 (See Appendix C), that function $h(\boldsymbol{x})$ is convex when $g(\boldsymbol{x})$ are those invex functions in Table 2.
> * **Q2.** Thanks for rising this up around Section 4.2. For compressed sensing, it is common to assume that the matrix $\boldsymbol{H}$ to contain both the image acquisition matrix $\boldsymbol{\Phi}$ and the sparse representation matrix $\boldsymbol{\Psi}$ (e.g., [66],[69],[70]). Given this setting, the conclusions we derive in Section 4.2 applies to all sparse signals in a given basis, and not just to canonical sparse vectors [69]. We have explained this in more detail in the new version of the paper to avoid any further ambiguity, and specifying also how images can be sparse. Please see lines 173, Section 4.2, and see references [61,66,69, 70].
> * **Q3.** The plug-and-play method studied in this paper is summarized in Algorithm 3 (Appendix G), which is a modified version of the APG method (Algorithm 1) and performs two proximal steps in Lines 5 and 6, respectively. Therefore, we incorporated the invex regularizers in Algorithm 3, by retaining Line 5 as in Algorithm 1, which explicitly uses an invex regularizer, and by replacing Line 6 with the Noise2Void CNN-based denoiser in Eq. (13). We have covered this in Section 4.2.2. With these arguments in place, we believe it is clear how we incorporated invex regularizers in a plug-and-play methodology. We have also modified the paper to provide a clear description around this. More specifically, we modified the description of Algorithm 3 in Section 4.2.2 (see Line 229-233). We also modified the text on Experiment 2 (Section 5.1), (Lines 314-317).
> * **Q4.** We believe we have now addressed this issue in the new version of the manuscript. We are now reporting additional results using TVAL3, ReconNet, and ISTA-Net, including SSIM as a metric as suggested by the reviewer. These results are summarized in Table 3 in the manuscript (in terms of PSNR) and in Tables 5, 6 and 7 of Appendix I (in terms of SSIM) of the supplemental material.
> * **Q5.** We have addressed this issue by providing a number of example reconstructed images in Appendices I and J, covering experiments 1, 2 and 3 (in Appendix I), and denoising (Appending J).
> # Discussion about weaknesses
> * 1. Apologies if there has been any confusion around our evaluation. The evaluation is not limited to FISTA (2010) and LISTA. It is worth highlighting that the LISTA version we are using for our evaluation is based on 2018 and not 2010 (as mentioned by the reviewer). Our baselines include BM3D, Accelerated Proximal Gradient method (APG), Plug-and-play version of APG with Noise2Void (2019), FISTA, LISTA, and five different invex regularizers (for denoising, and deconvolution problems). We opted for the BM3D, as it is one of the most successful and up to date denoisers that utilises the $\ell_{1}$-norm, and we firmly believe that is a suitable case to compare against the invex regularizers. The APG is a modern extended version of FISTA, highly utilised in a number of contemporary practical applications [55, 74, 75, 76].  As such, it is wrong to conclude that our evaluation is limited just to FISTA and LISTA, and we believe the results account for latest developments around deep learning-based methods like Noise2Void.
> * 2. The $\ell_{p}$-norm has, indeed, been included in the comparison, however, under the name of $\ell_{p}$-quasinorm (Equation (6)), covering both theoretical and numerical analysis, with $\ell_{p}$-quasinorm showing the best performance. In the revised draft, we are now reporting additional results for Experiment 1 by comparing invex regularizers, $\ell_{1}$-norm, SCAD, and Minimax-Concave regularizers. These results are reported in Table 3 of the manuscript (see Lines 311-312).
> * 3. We thank you for the suggested algorithms for comparison. In light of previous comment, and following your advice, we have complemented the compressive sensing numerical experiments by reporting additional results using TVAL3, ReconNet, and ISTA-Net. These results are summarized in Table 3 in the manuscript (in terms of PSNR) and in Tables 5, 6 and 7 of Appendix I (in terms of SSIM) of supplemental material.
> * 4. This issue has been addressed. See above.
> * 5. We have now complemented the current experimental results with examples of reconstructed images (along with their the reconstruction quality), reported in Appendices I (for experiments 1,2, and 3) and J (for denoising experiment) of supplemental material (owing to the reasons of space).

---

> > ### Comment · Reviewer_8pbb · 2022-08-07
> > **Reviewer 8pbb response to authors**
> >
> > I have revised my decision and review of the Paper3485.
> >
> > Reasons for the original decision:
> >
> > 1. Before this revision, the paper lacked comparisons with essential benchmarks. It was not an acceptable state.
> > 2. The paper focuses on computational imaging as an application, but only one image reconstruction example was provided.
> >
> > Reasons for the revised decision:
> >
> > 1. Authors responded to the technical inquiries with clarifications.
> > 2. Authors added comparisons with suggested benchmark methods in the case of image restoration tasks.
> > 3. Authors added image reconstruction examples as per suggestion.
> >
> > Minor issue: The revised paper is more than ten pages long. I believe it has to be limited to 9 pages until acceptance.

---

> > > ### Author Response · Authors · 2022-08-07
> > > **Response to reviewer 8pbb**
> > >
> > > We are glad to know that the reviewer found our rebuttal helpful. Thank you for the comments, and the support reconsidering the decision. Here we address your new questions.
> > > # Questions
> > > - **Q1.** We thank you for the suggested plot. We will include this new comparative plot between invex and convex regularizers in supplemental material of the final version of the manuscript.
> > > - **Q2.** Thanks for rising this up. We highlight that the intuition behind the advantages of the studied invex regularizers are attributed to the following reasons. The superiority of the $\ell_{p}$-quasinorm comes from the possibility of adjusting the value of $p$ in a data-dependent manner [49]. In the case of Equations (7), (8) , (9) and (10), their performance can be justified under the framework of reweighted $\ell_{1}$-norm minimization (please see [20] as an example), which enhances the performance of just $\ell_{1}$-norm minimization. Therefore, following your suggestion we will add a new subsection in Appendix A of supplemental material providing this discussion and more insights on the advantages of invex regularizers.
> > > - **Q3.** Following reviewer suggestion, we will add plots in the supplemental material of the final manuscript, numerically verifying the theoretical convergence results of this paper. These plots will show MSE vs iterations, and MSE vs time.
> > > - **Q4.** In light of the previous question, we will complement our analysis on the studied invex regularizers, by comparing their running time with convex regularizers under the analyzed image restoration methodologies.
> > > ## Minor Issue
> > > - Thanks for pointing this out. We highlight that our original manuscript had 9 pages, and the revised manuscript presented in the submission, is only intended to illustrate how we will address all comments of the reviewers, if our paper is accepted. Reason we marked all changes in blue.

---

### Meta-Review · Area_Chair_vyQW · 2022-08-25

**Recommendation:** Accept
**Confidence:** Certain

**Metareview:**

This paper addresses image reconstruction problems exploiting invex regularizers (which are not necessarily convex). For many modern signal processing applications, invexity of the cost is proved. Many examples are considered, and an extensive comparison with state-of-the art methods are provided in an application section.
As all reviewers point out unanimously, the paper is very well presented, original and overall of very high quality.  It is therefore a clear accept. But I recall the authors the strict rule on the number of pages for the final version (10 pages).

**Award:**

No

---

### Decision · Program_Chairs · 2022-09-14

Accept